# Come Together, But Not Right Now:
# A Progressive Strategy to Boost Low-Rank Adaptation

**Zhan Zhuang** [1 2]   **Xiequn Wang** [1]   **Wei Li** [1]   **Yulong Zhang** [3]   **Qiushi Huang** [1 4]   **Shuhao Chen** [1]
**Xuehao Wang** [1]   **Yanbin Wei** [1 5]   **Yuhe Nie** [6]   **Kede Ma** [2]   **Yu Zhang** [1]   **Ying Wei** [3]

## Abstract

Low-rank adaptation (LoRA) has emerged as a leading parameter-efficient fine-tuning technique for adapting large foundation models, yet it often locks adapters into suboptimal minima near their initialization. This hampers model generalization and limits downstream operators such as adapter merging and pruning. Here, we propose CoTo[1], a progressive training strategy that gradually increases adapters' activation probability over the course of fine-tuning. By stochastically deactivating adapters, CoTo encourages more balanced optimization and broader exploration of the loss landscape. We provide a theoretical analysis showing that CoTo promotes layer-wise dropout stability and linear mode connectivity, and we adopt a cooperative-game approach to quantify each adapter's marginal contribution. Extensive experiments demonstrate that CoTo consistently boosts single-task performance, enhances multi-task merging accuracy, improves pruning robustness, and reduces training overhead, all while remaining compatible with diverse LoRA variants. Code is available at https://github.com/zwebzone/coto.

## 1. Introduction

Parameter-efficient fine-tuning (PEFT) has become the dominant paradigm for adapting large foundation models (Radford et al., 2021; Rombach et al., 2022; Meta, 2024) to

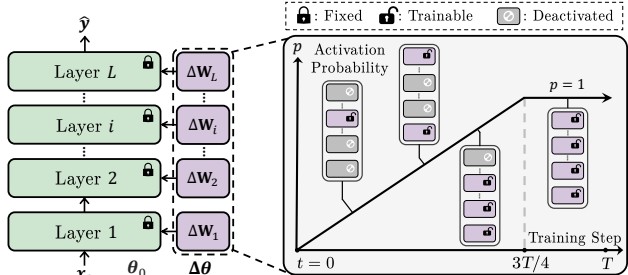

*Figure 1.* Illustration of the CoTo progressive activation schedule for LoRA adapters. For the first 75% of training (*i.e.*, $t < 3T/4$), each adapter is stochastically deactivated (shown in gray), where the activation probability $p(t)$ increases linearly from 0 to 1 as training progresses. In the final 25% of training, $p(t) = 1$, and all adapters remain active, reducing to full fine-tuning.

downstream tasks. By introducing a small number of trainable parameters, such as prompts (Lester et al., 2021), adapters (Houlsby et al., 2019), or low-rank adaptation (LoRA) modules (Hu et al., 2022), PEFT methods achieve rapid convergence and minimal storage overhead compared to full fine-tuning. Among these, LoRA has emerged as a particularly effective approach, reparameterizing weight updates as low-rank matrices[2].

Despite its success, vanilla LoRA often converges to suboptimal minima near their initialization, due to the "lazy" dynamics of standard gradient-based optimization (Du et al., 2018; Chizat et al., 2019). Moreover, empirical studies show a pronounced layer-wise imbalance (Dauphin et al., 2014): adapters in higher layers receive the bulk of the gradient signal and dominate task performance, while those in lower layers remain largely under-utilized (Zhang et al., 2023b; Gao et al., 2024a). This uneven optimization not only restricts single-task generalization but also hampers downstream operations, such as adapter merging (Zhao et al., 2024b) and pruning (Han et al., 2015; Li et al., 2017).

To mitigate these issues, we propose CoTo, a simple progressive training strategy that gradually increases each adapter's activation probability during fine-tuning. Early in training,

[1]Southern University of Science and Technology, Shenzhen, China [2]City University of Hong Kong, Hong Kong SAR, China [3]Zhejiang University, Hangzhou, China [4]University of Surrey, Surrey, UK [5]Hong Kong University of Science and Technology, Hong Kong SAR, China [6]New York University, New York, USA. Correspondence to: Kede Ma <kede.ma@cityu.edu.hk>, Yu Zhang <yu.zhang.ust@gmail.com>, Ying Wei <ying.wei@zju.edu.cn>.

*Proceedings of the 42nd International Conference on Machine Learning*, Vancouver, Canada. PMLR 267, 2025. Copyright 2025 by the author(s).

[1]This acronym nods to the Beatles' classic song 'Come Together'—*but not* right now.

[2]Throughout this paper, we also call LoRA's trainable parameters adapters. Specifically, each adapter corresponds to all LoRA parameters within a single Transformer layer.

CoTo stochastically deactivates a random subset of adapters, forcing the model to distribute gradient updates more evenly, and then linearly raises the activation probability until all adapters participate fully. This curriculum-like scheme encourages broader exploration of the loss landscape, yields layer-wise dropout stability, and promotes linear mode connectivity (LMC) between independently trained solutions.

We provide a theoretical analysis showing that CoTo minimizes an upper bound on a weighted ensemble of sub-network losses and, via a cooperative-game perspective, quantifies each adapter's marginal contribution using Shapley values (Shapley, 1953). Empirically, CoTo consistently boosts single-task generalization, enhances multi-task adapter merging accuracy, improves adapter pruning robustness, and reduces overall training cost. Crucially, it requires no architectural changes, and integrates seamlessly with existing LoRA variants and advanced update schemes.

## 2. Related Work

**Parameter-Efficient Fine-Tuning.** The rapid growth of foundation models has spurred extensive research into PEFT techniques, aiming to adapt large pre-trained networks to downstream tasks without incurring the computational and storage costs of full fine-tuning. Early PEFT methods introduce modular components, such as adapters (Houlsby et al., 2019), prompts (Lester et al., 2021), or prefixes (Li & Liang, 2021), to capture task-specific knowledge while freezing the bulk of the pre-trained parameters. Houlsby et al. (2019) first proposed adapter layers (*i.e.*, small bottleneck modules) inserted into Transformer blocks. Prompt-tuning (Lester et al., 2021) and prefix-tuning (Li & Liang, 2021) similarly leverage learnable tokens or continuous prefixes to steer the model toward a new task.

LoRA (Hu et al., 2022), on the other hand, reframes fine-tuning as the problem of learning low-rank updates to each weight matrix. Instead of adding full-rank adapter layers, LoRA factorizes the weight update into two low-rank matrices and injects them into each Transformer layer. This decomposition dramatically reduces the number of trainable parameters while consistently delivering stronger performance than earlier PEFT methods. Subsequent work has proposed various LoRA extensions, with the goal of improving adaptation quality, reducing parameter count further, or aligning with full fine-tuning dynamics. For example, DoRA (Liu et al., 2024a) decomposes LoRA updates into magnitude and directional components to better approximate the full, high-dimensional updates, whereas HiRA (Huang et al., 2025) applies Hadamard products between low-rank matrices and the original weights to enable high-rank adaptation without significantly increasing parameter cost. Other notable variants include LoRA-FA (Zhang et al., 2023a), which freezes the projection-down weights for

greater stability, and FourierFT (Gao et al., 2024b), which leverages Fourier transforms to represent weight updates in the frequency domain. In parallel, adaptive rank schemes such as AdaLoRA (Zhang et al., 2023b), ALoRA (Liu et al., 2024b), and LoRA-drop (Zhou et al., 2024) automatically adjust the rank per layer. These variants underscore the flexibility of the low-rank paradigm but also highlight the persistent challenge of ensuring balanced, layer-wise utilization of adapters during optimization.

Beyond computational innovations, several studies have focused on improving the initialization and optimization dynamics of LoRA. PiSSA (Meng et al., 2024) uses truncated singular value decomposition of the pre-trained weights to initialize LoRA matrices. LoRA-GA (Wang et al., 2024b) aligns the LoRA initialization with gradient-based approximations of full fine-tuning, while rsLoRA (Kalajdzievski, 2023) adjusts scaling factors to stabilize early training. On the optimization side, LoRA+ (Hayou et al., 2024) employs distinct learning rates for the two low-rank matrices. Similarly, LoRA-Pro (Wang et al., 2024c) modifies the gradient updates to more closely emulate the behavior of full fine-tuning. While these methods yield improvements in convergence speed or final performance, they do not explicitly address the problem of layer-wise imbalance.

**Model Merging.** Combining task-specific adapters to form a single set of parameters that performs well on multiple tasks relies on the property of LMC (Frankle et al., 2020; Entezari et al., 2022; Zhou et al., 2023), which posits that two independently fine-tuned solutions often lie in loss basins connected by a low-loss linear path. In the LoRA context, LoraHub (Huang et al., 2023) first demonstrates that merging low-rank adapters trained on separate language tasks can yield models with strong generalization to new tasks. Federated learning extensions, such as FedIT (Zhang et al., 2024) and FLoRA (Wang et al., 2024d) apply LoRA merging and stacking across distributed clients, mitigating catastrophic forgetting and communication overhead. Recently, LoRA-LEGO (Zhao et al., 2024b) clusters semantically similar LoRA "units" within each layer before merging to reduce task interference, while ZipLoRA (Shah et al., 2025) focuses on disentangling style and content subspaces to enable compositional generation in diffusion models. Despite these advances, effective multi-task merging remains challenging when adapters converge to layer-wise imbalanced minima.

**Stochastic Regularization** methods, originally developed to prevent overfitting, have been adapted to the LoRA setting to encourage robustness and exploration of the parameter space. Classical techniques like Dropout (Srivastava et al., 2014) and DropConnect (Wan et al., 2013) randomly zero out elements or connections during training. Stochastic Depth (Huang et al., 2016) and LayerDrop (Fan et al., 2020) skip entire layers with a fixed or linearly decaying proba-

bility. Within LoRA, entry-wise or column-wise dropout has been explored (Wang et al., 2024a; Lin et al., 2024) to regularize low-rank matrices, but these approaches do not account for the sequential, layer-wise computation of adapters. Consequently, they may fail to correct the disproportionate updates received by higher-layer adapters. In contrast, the proposed CoTo introduces a progressive training strategy that dynamically increases the activation probability of each adapter early in training. This curriculum-like schedule balances gradient flow across all layers, fosters exploration of diverse subnetworks, and improves downstream operations.

## 3. Proposed Method: CoTo

In this section, we introduce the proposed CoTo and present two complementary perspectives to elucidate its behavior.

### 3.1. Preliminaries

Let the parameters of a pre-trained foundation model be $\boldsymbol{\theta}_0 = \{\mathbf{W}_i\}_{i=1}^L$, where $\mathbf{W}_i \in \mathbb{R}^{m \times n}$ is the weight matrix of layer $i$. For an input $\boldsymbol{x}_0$, the model $f = g \circ h_L \circ \cdots \circ h_2 \circ h_1$ computes a sequence of hidden features: $\boldsymbol{x}_i = h_i(\boldsymbol{x}_{i-1}, \mathbf{W}_i)$ for $i \in \{1, 2 \ldots, L\}$, and produces the final output with the prediction head $g$: $\hat{\boldsymbol{y}} = g(\boldsymbol{x}_L)$. In LoRA, we freeze each base weight $\mathbf{W}_i$ and introduce a low-rank update $\Delta \mathbf{W}_i$. Concretely, LoRA factorizes each update as $\Delta \mathbf{W}_i = \alpha \mathbf{BA}$, where $\mathbf{A} \in \mathbb{R}^{r \times n}$, $\mathbf{B} \in \mathbb{R}^{m \times r}$, and $r \ll \min(m, n)$ controls the rank. The scaling factor $\alpha$ adjusts the magnitude of the update.

### 3.2. Training Strategy

CoTo, as illustrated in Figure 1, introduces a simple, progressive schedule for stochastically deactivating adapters during the early stages of fine-tuning, and then gradually "turning them on" so that, by the final stages, all adapters participate fully. Specifically, for each layer $i$, we draw an activation indicator:

$$\delta_i \sim \text{Bernoulli}(p(t)), \tag{1}$$

where $p(t) \in [0, 1]$ is a time-dependent probability that increases linearly from 0 to 1 over the first 75% of training steps, and remains equal to 1 for the remaining 25%. Denoting the total number of training steps by $T$, at step $t \in \{1, \ldots, T\}$, we set

$$p(t) = \begin{cases} \dfrac{4t}{3T} & t < \dfrac{3T}{4} \\ 1 & t \geq \dfrac{3T}{4}. \end{cases} \tag{2}$$

Accordingly, the model output is adjusted to

$$\hat{\boldsymbol{y}} = f\left(\boldsymbol{x}_0; \{\mathbf{W}_i + \delta_i \mathbf{1} \odot \Delta \mathbf{W}_i\}_{i=1}^L\right), \tag{3}$$

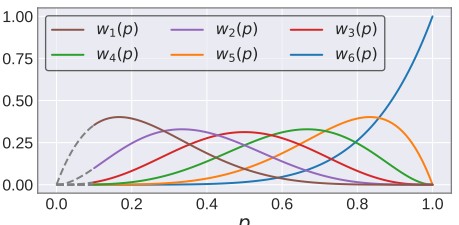

*Figure 2.* Visualization of the weight function $w_j(p)$ in Eq. (6).

where $\mathbf{1}$ is an all-ones matrix of the same size as $\mathbf{W}_i$, and $\odot$ denotes the Hadamard product. The training objective is to minimize the expected loss:

$$\min_{\{\Delta \mathbf{W}_i\}} \mathbb{E}_{\boldsymbol{\delta}}\left[\ell(\hat{\boldsymbol{y}}, \boldsymbol{y})\right], \tag{4}$$

where $\boldsymbol{\delta} = [\delta_1, \ldots, \delta_L]^\mathsf{T} \in \{0, 1\}^L$, $\boldsymbol{y}$ is the target label, and $\ell$ is typically cross-entropy loss for classification or mean squared error for regression.

### 3.3. Training Dynamics

**Curriculum of Subnetworks.** Early in training, when the probability $p$ of activating adapters is low, only a few adapters participate, forcing gradient updates to spread across layers and preventing higher-layer adapters from dominating the loss signal; as $p$ increases, more adapters are gradually engaged, expanding the space in which the model can fine-tune. This stochastic deactivation also counters the "lazy" regime—where gradients tend to stay near initialization—by encouraging exploration of a broader parameter region. By the time all adapters are active, the model has already diversified its search and is less likely to converge to poor, layer-imbalanced minima.

**Computational Savings.** Whenever $\delta_i = 0$, adapter $i$ is skipped entirely—no matrix multiplications involving $\mathbf{A}_i$ or $\mathbf{B}_i$ are performed. Thus, CoTo reduces both forward and backward computation in the early training stages.

### 3.4. Progressive Optimization Perspective

In this subsection, we view CoTo as training a weighted ensemble of partial LoRA configurations (*i.e.*, subnetworks that omit certain adapters). This perspective makes precise how CoTo encourages both robustness to adapter dropout and improved connectivity between different minima.

Specifically, denote by

$$\tilde{\boldsymbol{y}}_j = \mathbb{E}_{\|\boldsymbol{\delta}\|_1 = j}\left[f\left(\boldsymbol{x}_0; \{\mathbf{W}_i + \delta_i \mathbf{1} \odot \Delta \mathbf{W}_i\}_{i=1}^L\right)\right], \tag{5}$$

the expected model prediction over all subsets of adapters of size $j$, where $\|\boldsymbol{\delta}\|_1 = \sum_{i=1}^L \delta_i$. When $j = L$, all adapters are active and $\tilde{\boldsymbol{y}}_L$ recovers the standard LoRA model output.

*Table 1.* Average accuracy (%) on 11 image classification tasks. The highest accuracy (%) is **bolded**, while the second highest is underlined. CLIP results are copied from (Zanella & Ben Ayed, 2024). All adapters use a rank of $r = 2$ with ViT-B/16 as the backbone.

| Method | Aircraft | Caltech | Cars | DTD | EuroSAT | Flowers | Food | ImageNet | Pets | SUN | UCF | Avg |
|---|---|---|---|---|---|---|---|---|---|---|---|---|
| CLIP | 24.7 | 92.9 | 65.3 | 43.6 | 47.5 | 71.4 | 86.1 | 66.7 | 89.1 | 62.6 | 66.7 | 65.1 |
| LoRA (ICLR'22) | 53.89 | 96.25 | 85.12 | 72.00 | 92.05 | 97.78 | 85.15 | 73.49 | 93.27 | 76.75 | 86.72 | 82.95 |
| LoRA-CoTo | 55.69 | 96.26 | 86.04 | 72.68 | 92.97 | 98.12 | 85.48 | 73.53 | 93.42 | 76.85 | 87.20 | 83.48 |
| DoRA (ICML'24) | 56.21 | 96.38 | 86.67 | 72.60 | 92.13 | 98.09 | 85.04 | 73.54 | 93.59 | 76.54 | 87.15 | 83.45 |
| DoRA-CoTo | 57.35 | 96.51 | 86.98 | 72.83 | 93.45 | 98.25 | 86.31 | 73.62 | 94.06 | 76.90 | 86.99 | 83.93 |
| HiRA (ICLR'25) | 57.62 | 96.35 | 87.22 | 73.38 | 92.51 | 98.06 | 86.72 | 73.76 | 94.41 | 76.92 | 86.81 | 83.98 |
| HiRA-CoTo | **57.85** | **96.65** | **87.40** | **73.71** | **93.46** | **98.71** | **86.91** | **73.85** | **94.46** | **77.36** | **87.38** | **84.34** |

At iteration $t$, CoTo samples a random vector $\boldsymbol{\delta}$, where each $\delta_i$ is Bernoulli($p(t)$). Over this randomness, the probability that exactly $j$ adapters are active is

$$w_j\big(p(t)\big) = \binom{L}{j} p(t)^j \big(1 - p(t)\big)^{L-j}, j = 0, \dots, L, \quad (6)$$

as illustrated in Figure 2.

**Theorem 3.1.** *Let $\ell(\cdot, \boldsymbol{y})$ be a convex loss function. Then for any fixed $p \in [0, 1]$,*

$$\min_{\{\Delta \mathbf{W}_i\}} \mathbb{E}_{\boldsymbol{\delta}} \left[ \ell\left(\hat{\boldsymbol{y}}, \boldsymbol{y}\right) \right] \geq \min_{\{\Delta \mathbf{W}_i\}} \sum_{j=1}^{L} w_j(p) \, \ell\left(\tilde{\boldsymbol{y}}_j, \boldsymbol{y}\right).$$

*Consequently, the expected CoTo objective at step $t$ upper-bounds a binomially weighted sum of the subnetwork losses.*

Theorem 3.1 follows directly from applying Jensen's inequality to the convex loss $\ell(\cdot; \boldsymbol{y})$. A detailed proof is given in Appendix D.

Because CoTo's training objective accounts for all possible choices of active adapters (weighted by $w_j(p)$), the model is explicitly encouraged to perform well even if any subset of adapters is disabled. Prior work (Frankle et al., 2020; Adilova et al., 2024) shows that dropout stability often implies that independently trained solutions can be connected by a low-loss linear path. Intuitively, because CoTo trains adapters in near-isolation (for low $p$) before gradually re-enabling them, each adapter's parameters learn a solution "locally," reducing inter-adapter dependencies. Consequently, two CoTo-trained models with different random seeds tend to lie in loss valleys that are linearly connected. Empirical verification appears in Section 4.2.

### 3.5. Cooperative-Game Perspective

An alternative way to understand CoTo is through the lens of a *cooperative game*: each adapter is treated as a "player" in a game whose "value function" is the model performance when that subset of adapters is active. By attributing the

*marginal contribution* of each adapter to overall performance, we identify precisely how CoTo encourages balanced layer-wise optimization.

Let $\mathcal{S} = \{1, \dots, L\}$ index the set of $L$ adapters (one per player). For any subset $\mathcal{R} \subset \mathcal{S}$, define the value function:

$$v(\mathcal{R}) = \mathbb{E}_{\boldsymbol{x}} \left[ \ell \left( f \left( \boldsymbol{x}; \{ \mathbf{W}_i + \delta_i \mathbf{1} \odot \Delta \mathbf{W}_i \}_{i=1}^{L} \right), \boldsymbol{y} \right) \right], (7)$$

where $\delta_i = 1$ if $i \in \mathcal{R}$ and 0 otherwise. Under this interpretation, the Shapley value (Shapley, 1953) of adapter $i$ can be approximated efficiently using the multilinear extension approach (Owen, 1972):

$$\varphi_i(v) = \int_0^1 c_i(p) dp, \ c_i(p) = \mathbb{E}\left[ v\left(\mathcal{R}_i \cup \{i\}\right) - v\left(\mathcal{R}_i\right) \right],$$

where $\mathcal{R}_i$ indexes a random subset of adapters excluding adapter $i$, and $c_i(p)$ captures the expected *marginal contribution* of adapter $i$ when selected with probability $p$. Therefore, to estimate $\varphi_i(v)$, one may sample a few values of $p$, draw random subsets $\mathcal{R}_i$, compute the difference $v\left(\mathcal{R}_i \cup \{i\}\right) - v\left(\mathcal{R}_i\right)$ (again approximated by averages over a finite set of samples in Eq. (7)), and average appropriately. By inspecting $\varphi_i(v)$ after CoTo training, we gain insights into each adapter's marginal contribution.

## 4. Experiments

We evaluate CoTo's effectiveness through a series of experiments designed to answer three key questions: 1) Can CoTo improve single-task generalization across diverse benchmarks? 2) Does CoTo facilitate LMC for more effective model merging? 3) Can CoTo enhance pruning robustness? All experiments use three random seeds to ensure statistical reliability, and implementation details are deferred to Appendix A.

### 4.1. Single-Task Generalization

**Results on Vision Benchmarks.** To assess CoTo's impact in the vision domain, we follow the CLIP-LoRA

*Table 2.* Average accuracy (%) on 8 commonsense reasoning tasks (Hu et al., 2023) using LLaMA-2-7B and LLaMA-3-8B backbones. All adapters use a rank of $r = 32$. Results without CoTo are copied from (Huang et al., 2025).

| Model | Method | ARC-c | ARC-e | BoolQ | HellaS | OBQA | PIQA | SIQA | WinoG | Avg |
|---|---|---|---|---|---|---|---|---|---|---|
| ChatGPT | N/A | 79.90 | 89.80 | 73.10 | 78.50 | 74.80 | 85.40 | 68.50 | 66.10 | 77.01 |
| LLaMA-2-7B | LoRA | 64.70 | 79.80 | 69.80 | 83.60 | 81.00 | 79.90 | 79.50 | 82.60 | 77.61 |
| | LoRA-CoTo | 69.58 | 85.33 | 71.48 | **91.15** | 82.10 | 82.89 | 78.94 | 83.54 | 80.63 |
| | DoRA | 68.20 | 83.70 | 71.80 | 89.10 | 82.40 | 83.70 | 76.00 | 82.60 | 79.69 |
| | DoRA-CoTo | 69.51 | 85.08 | **72.25** | 90.82 | 81.33 | 83.10 | 79.50 | 83.43 | 80.64 |
| | HiRA | 73.81 | 86.74 | 71.22 | 88.12 | **84.60** | 83.35 | 79.53 | 83.98 | 81.42 |
| | HiRA-CoTo | **74.49** | **87.08** | 72.11 | 88.40 | 84.00 | **84.33** | 79.89 | **85.24** | **81.94** |
| LLaMA-3-8B | LoRA | 71.20 | 84.20 | 70.80 | 91.70 | 79.00 | 85.20 | 79.90 | 84.30 | 80.79 |
| | LoRA-CoTo | 79.35 | 90.81 | 75.02 | 94.77 | 85.20 | 88.39 | 80.55 | 86.08 | 85.02 |
| | DoRA | 80.40 | 90.50 | 74.60 | 95.50 | 85.80 | 89.30 | 79.90 | 85.60 | 85.20 |
| | DoRA-CoTo | 79.38 | 91.50 | **75.40** | 95.98 | 86.00 | 88.52 | 81.12 | 86.00 | 85.49 |
| | HiRA | 82.90 | **93.27** | **75.40** | 95.36 | 88.32 | 89.70 | 81.15 | 87.70 | 86.72 |
| | HiRA-CoTo | **83.36** | **93.27** | 75.32 | 95.42 | **88.40** | **90.15** | **81.99** | **88.08** | **87.00** |

*Table 3.* Average accuracy (%) on mathematical reasoning tasks (Cobbe et al., 2021) using the LLaMA-2-7B backbone. All adapters use a rank of $r = 8$. Results without CoTo are copied from (Wang et al., 2024c).

| Method | Params (%) | w/o CoTo | w/ CoTo |
|---|---|---|---|
| LoRA | 0.296 | 42.08 ± 0.04 | 55.85 ± 0.74 |
| DoRA | 0.316 | 53.07 ± 0.75 | 56.56 ± 0.19 |
| HiRA | 0.296 | 54.51 ± 0.59 | 56.68 ± 0.09 |
| PiSSA | 0.296 | 44.54 ± 0.27 | 50.16 ± 0.47 |
| rsLoRA | 0.296 | 45.62 ± 0.10 | 56.99 ± 0.66 |
| LoRA+ | 0.296 | 52.11 ± 0.62 | 54.36 ± 0.43 |
| LoRA-Pro | 0.296 | 54.23 ± 0.79 | 57.16 ± 0.38 |

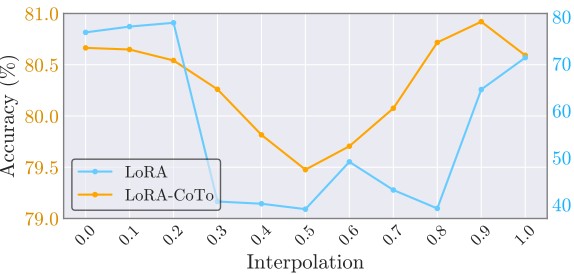

*Figure 3.* Linear interpolation accuracy on commonsense reasoning tasks (Hu et al., 2023). CoTo's interpolation curve averaged across 8 tasks (orange) remains flatter and higher compared to vanilla LoRA (blue), demonstrating superior LMC.

setup (Zanella & Ben Ayed, 2024), fine-tuning the ViT-B/16 backbone on 11 image classification datasets assembled by Zhou et al. (2022). Each dataset defines an independent few-shot task with 16 training images per class. We compare three LoRA variants—vanilla LoRA, DoRA (Liu et al., 2024a), and HiRA (Huang et al., 2025)—both with and without CoTo. Table 1 reports average accuracies over three seeds. CoTo yields noticeable performance gains across all LoRA variants, demonstrating that progressive training leads to more balanced utilization of all adapters.

**Results on Language Benchmarks.** In the language domain, we first evaluate CoTo on commonsense reasoning tasks using LLaMA backbones (Meta, 2023; 2024). Following Wang et al. (2024c), we fine-tune LLaMA-2-7B and LLaMA-3-8B on the Commonsense170K suite (Hu et al., 2023), which comprises 8 tasks. All adapter variants use a rank of $r = 32$. Table 2 reports average accuracies, from which we observe consistent performance improvements across different backbones, LoRA variants, and task complexities. These results indicate that CoTo's progressive activation schedule helps mitigate layer-wise imbalance and

lazy convergence, especially as model capacity increases.

We further evaluate CoTo on mathematical reasoning tasks by fine-tuning LLaMA-2-7B on MetaMathQA (Yu et al., 2024) and testing it on GSM8K (Cobbe et al., 2021). In this setting, we adjust the adapter rank to 8 and compare CoTo against PiSSA (Meng et al., 2024), rsLoRA (Kalajdzievski, 2023), LoRA+ (Hayou et al., 2024), and LoRA-Pro (Wang et al., 2024c). Similar performance gains have been achieved, as shown in Table 3.

### 4.2. Single-Task and Multi-Task Model Merging

In this subsection, we assess the LMC property through two complementary experiments.

**Single-Task Model Merging.** To quantify how well two independently trained LoRA solutions can be connected by a linear path, we fine-tune two instances on the same task from different random seeds, and then linearly interpolate their adapter parameters for interpolation ratios $\lambda \in [0, 1]$. Figure 3 illustrates interpolation accuracies on commonsense reasoning tasks (Hu et al., 2023): at $\lambda = 0.5$ (equal mix-

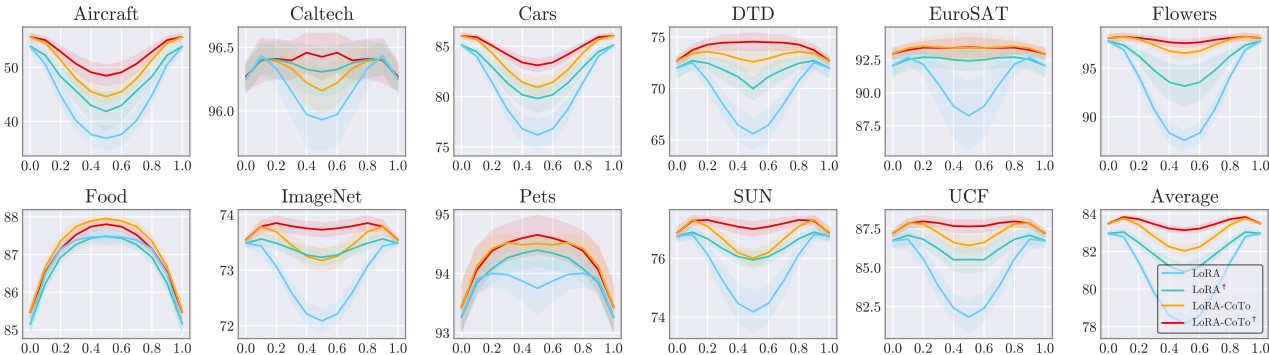

*Figure 4.* Linear interpolation accuracy on 11 image classification tasks. CoTo (orange) consistently outperforms vanilla LoRA (blue). Adding alignment (denoted by $^\dagger$) improves both but preserves CoTo's margin.

ture), LoRA-CoTo maintains 79% accuracy, whereas vanilla LoRA drops to 39%. Across the entire interpolation trajectory, CoTo's curve remains substantially flatter and higher, indicating that independently CoTo-trained adapters lie in closer, low-loss basins. Analogous trends emerge in the image classification experiments (see Figure 4). Even after applying an additional "alignment" step: learning an invertible matrix $\mathbf{P}$ to minimize $\|\Delta\mathbf{W}_f - \Delta\mathbf{W}_e\|_2$, where $\Delta\mathbf{W}_f$ and $\Delta\mathbf{W}_e$ denote linear weight fusion and model ensemble, respectively, CoTo retains its advantages over vanilla LoRA[3]. Analysis of $\|\Delta\mathbf{W}_f - \Delta\mathbf{W}_e\|_2$ and $\|\mathbf{P}\|_2$ in Figure 5 further confirms that CoTo's performance gains stem from balanced layer-wise optimization rather than mere post-hoc alignment.

**Multi-Task Model Merging.** Building on CoTo's enhanced LMC, we next examine its impact on merging adapters trained on different tasks following the experimental setup and default configurations in (Zhao et al., 2024b). First, we consider multi-task merging for generative language understanding tasks (Wang et al., 2019). We test on seven in-domain tasks and two out-of-domain tasks (Longpre et al., 2023) via the same prompt format (Wei et al., 2022). Both LLaMA-2-7B and LLaMA-2-13B backbones are used. We employ three merging strategies: linear weight fusion (*i.e.*, $\Delta\mathbf{W}_f$), linear model ensemble (*i.e.*, $\Delta\mathbf{W}_e$), and the LoRA-LEGO method proposed by Zhao et al. (2024b), which explicitly aligns and fuses parameter updates. As reported in Table 4, CoTo-trained adapters yield markedly better merging performance. On LLaMA-2-7B, linear weight fusion of CoTo-based adapters improves average accuracy from 47.17% to 58.53% (+11.36%), and LoRA-LEGO merging rises from 62.21% to 67.19% (+4.98%). The ensemble approach also benefits some tasks, though gains are more

---

[3]Linear weight fusion is computed by $\Delta\mathbf{W}_f = (\lambda\mathbf{B}_1 + (1-\lambda)\mathbf{B}_2)(\lambda\mathbf{A}_1 + (1-\lambda)\mathbf{A}_2)$, which preserves the rank, while linear model ensemble (Zhao et al., 2024a) is computed by $\Delta\mathbf{W}_e = \lambda\mathbf{B}_1\mathbf{A}_1 + (1-\lambda)\mathbf{B}_2\mathbf{A}_2$. To insert the learnable invertible matrix $\mathbf{P}$, we reparameterize $\Delta\mathbf{W}_2 = \mathbf{B}_2\mathbf{A}_2 = (\mathbf{B}_2\mathbf{P})(\mathbf{P}^{-1}\mathbf{A}_2)$.

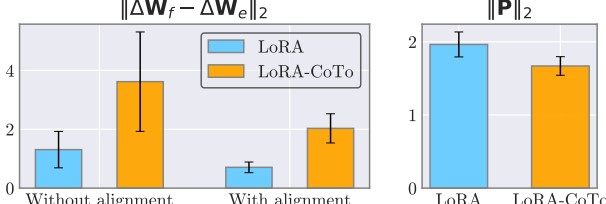

*Figure 5.* Analysis of the optimized alignment matrix $\mathbf{P}$ for LoRA$^\dagger$ and LoRA-CoTo$^\dagger$. Error bars denote standard deviations.

modest. Similar trends are observed for LLaMA-2-13B, indicating that CoTo encourages each adapter to converge to parameters that lie in closer, low-loss subspaces.

We also evaluate CoTo's efficacy for merging adapters on six discriminative language understanding tasks (Wang et al., 2019) using DeBERTa-v3 (He et al., 2023). Although merging classifiers across distinct tasks is inherently more challenging due to differences in feature pooling and output prediction, CoTo-trained adapters still exhibit consistent improvements across all three merging strategies (see Table 11 in the Appendix). These findings indicate that CoTo is compatible with existing merging techniques and consistently enhances multi-task LoRA merging across both generative and discriminative architectures.

Finally, we explore CoTo in the context of diffusion-based generative models. Using SDXL (Podell et al., 2024) as our backbone, we apply DreamBooth (Ruiz et al., 2023) to fine-tune separate style and object adapters—specifically, object LoRAs for two categories (*i.e.*, cat and dog) and style LoRAs for two artistic styles (*i.e.*, watercolor and crayon). We then merge style and object adapters using ZipLoRA (Shah et al., 2025). Qualitative results in Figure 6 demonstrate that CoTo significantly reduces style and object forgetting: for instance, a crayon-style cat generated with LoRA-CoTo clearly preserves both the cat's identity and "crayon-ness," whereas vanilla LoRA often compromises one or the other. This qualitative evidence further attests to

*Table 4.* Average accuracy (%) on multi-task merging for 9 generative language understanding tasks (Wang et al., 2019) using LLaMA-2-7B and LLaMA-2-13B backbones. "Fusion" and "Ensemble" correspond to the linear weight fusion and linear model ensemble that compute $\Delta\mathbf{W}_f$ and $\Delta\mathbf{W}_e$, respectively. Results without CoTo are copied from (Zhao et al., 2024b).

| Model | Method | In-Domain Task | | | | | | | Out-of-Domain Task | | Avg |
| --- | --- | --- | --- | --- | --- | --- | --- | --- | --- | --- | --- |
| | | CoLA | MNLI | MRPC | QNLI | QQP | RTE | SST2 | SNLI | WNLI | |
| LLaMA-2-7B | Task-Specific LoRA | 61.63 | 77.46 | 68.00 | 77.25 | 75.83 | 52.22 | 75.74 | – | – | – |
| | Fusion | 54.42 | 36.09 | 68.00 | 44.41 | 51.72 | 48.15 | 42.99 | 31.64 | 47.14 | 47.17 |
| | Fusion-CoTo | **57.31** | 47.39 | 61.75 | 62.60 | 71.89 | 71.11 | 60.80 | 36.79 | 57.14 | 58.53 |
| | Ensemble | 55.67 | 45.89 | 59.25 | 59.84 | 67.38 | 68.89 | 66.44 | 36.73 | 51.43 | 56.84 |
| | Ensemble-CoTo | 57.21 | 45.68 | 47.75 | 61.39 | 68.59 | 69.26 | 60.57 | 35.24 | **64.29** | 56.66 |
| | LoRA-LEGO | 55.48 | 55.73 | 66.00 | 62.29 | 71.07 | **71.85** | 73.22 | 51.36 | 52.86 | 62.21 |
| | LoRA-LEGO-CoTo | 53.94 | **64.35** | 72.25 | 72.71 | 78.51 | 71.48 | 75.75 | 58.59 | 57.14 | **67.19** |
| LLaMA-2-13B | Task-Specific LoRA | 69.04 | 88.23 | 89.25 | 82.33 | 86.29 | 80.74 | 76.44 | – | – | – |
| | Fusion | 45.48 | 46.32 | 67.75 | 46.68 | 47.50 | 62.96 | 46.78 | 42.42 | 42.86 | 49.86 |
| | Fusion-CoTo | **64.52** | 57.82 | 73.75 | 66.10 | 78.53 | 75.93 | 75.52 | 42.28 | **67.14** | 66.84 |
| | Ensemble | 62.50 | 64.64 | 74.75 | 71.81 | 81.35 | **79.26** | 75.52 | 54.32 | 60.00 | 69.35 |
| | Ensemble-CoTo | 63.75 | 60.54 | 71.75 | 67.82 | 76.38 | 77.78 | 75.75 | 46.79 | 62.86 | 67.05 |
| | LoRA-LEGO | 59.42 | 65.40 | 75.50 | 72.29 | 82.51 | 78.52 | 75.98 | 58.54 | 64.29 | 70.27 |
| | LoRA-LEGO-CoTo | 61.83 | **65.75** | **78.25** | 76.81 | **82.90** | 77.78 | **76.32** | 58.74 | 65.71 | **71.57** |

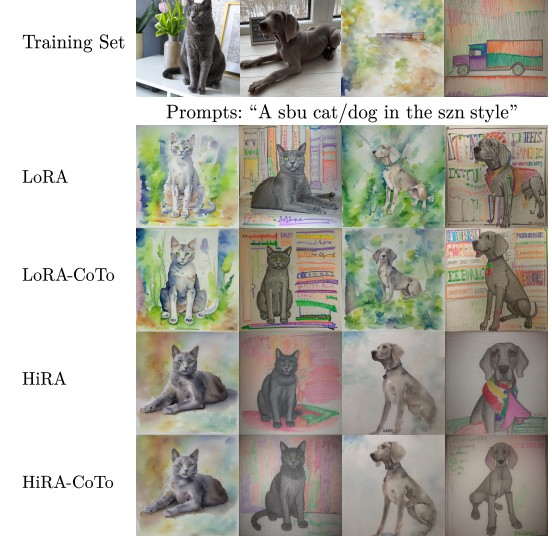

*Figure 6.* Customized sample images generated by SDXL (Podell et al., 2024) with and without CoTo. When merging style and object adapters via ZipLoRA (Shah et al., 2025), CoTo preserves both the object identity and artistic style more faithfully than vanilla LoRA. Each comparison uses the same seed.

CoTo's ability to learn adapters that merge more coherently across diverse vision and language tasks.

### 4.3. Model Pruning

The stochastic nature of CoTo naturally lends itself to improved pruning robustness, as adapters are trained to maintain performance even when a random subset is deactivated. To systematically evaluate this property, we conduct both

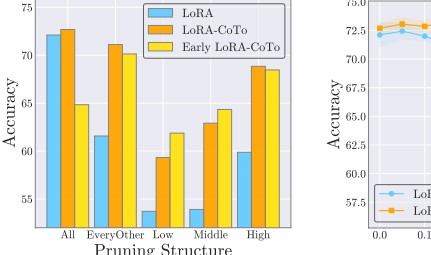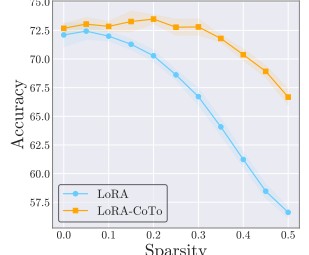

*Figure 7.* Average accuracy (%) on model pruning for the visual texture classification task (Cimpoi et al., 2014). Left panel: Structured pruning applied to LoRA, LoRA-CoTo, and Early LoRA-CoTo under four pruning patterns: alternating layers (EveryOther), first 4 layers (Low), middle 4 layers (Middle), and last 4 layers (High). "All" denotes unpruned models. Right panel: Unstructured pruning with varying sparsity.

structured and unstructured pruning experiments. We first examine layer-wise structured pruning on the visual texture classification task (Cimpoi et al., 2014) by selectively removing adapters from different network layers. As shown in the left panel of Figure 7, we compare four configurations: 1) removing alternating layers (denoted by EveryOther), 2) pruning the first 4 layers (denoted by Low), 3) pruning middle 4 layers (denoted by Middle), and 4) pruning the last 4 layers (denoted by High). The results demonstrate that CoTo-trained adapters maintain significantly better performance across all pruning patterns compared to vanilla LoRA. Notably, the "Early LoRA-CoTo" checkpoint (sampled right after the first 25% of training) already shows strong pruning robustness, indicating that the benefits emerge early in the progressive training schedule. Complete results are detailed in Figure 12 of the Appendix.

Table 5. Mean Euclidean distance between LoRA adapter weights across four learning rates. Init. to Final: Distance from initial to final weights. Final to Final: Distance between final weights from different initialization. Note that the mean initial distances are 1.155 for independent random seeds and 0.02 for perturbed seeds.

| Distance | 5e-5 | 1e-4 | 5e-4 | 1e-3 |
|---|---|---|---|---|
| Random Initialization | | | | |
| Init. vs. Final (w/o CoTo) | 0.48 | 0.45 | 0.76 | 1.32 |
| Init. vs. Final (w/ CoTo) | 0.61 | 0.79 | 1.25 | 1.64 |
| Final vs. Final (w/o CoTo) | 1.70 | 1.81 | 2.35 | 3.12 |
| Final vs. Final (w/ CoTo) | 1.38 | 1.53 | 2.14 | 2.63 |
| Same Initialization with Minor Additive Uniform Noise | | | | |
| Final vs. Final (w/o CoTo) | 0.05 | 0.07 | 0.59 | 1.71 |
| Final vs. Final (w/ CoTo) | 0.04 | 0.06 | 0.21 | 0.52 |

For fine-grained sparsity analysis, we evaluate unstructured pruning by zeroing out increasing percentages of adapter parameters. The right panel of Figure 7 shows that the performance gap widens with increasing sparsity level, and at 50% sparsity, LoRA-CoTo achieves 10% higher accuracy than vanilla LoRA. These findings highlight that CoTo enables more aggressive pruning while maintaining model utility even at high sparsity levels.

### 4.4. Further Analysis on DTD

**Convergence Near Initialization.** We empirically validate that LoRA adapters converge near their initialization, consistent with "lazy training" dynamics (Chizat et al., 2019). Using t-SNE visualization (Van der Maaten & Hinton, 2008) (see Figure 14 in the Appendix), we find that final adapter weights form tight clusters around their respective initialization points across five random seeds and four learning rates. This local convergence persists regardless of whether CoTo is applied. Nevertheless, CoTo yields slightly larger distances (see Table 5), suggesting broader exploration. Moreover, final weights from independent random seeds are closer under CoTo, indicating more consistent convergence paths. When initialized from the same point with minor additive uniform noise, CoTo-trained adapters converge to tighter clusters compared to vanilla LoRA, demonstrating robustness to initialization perturbations.

**Adapter Contribution Analysis.** To quantify layer-wise marginal utilization, we compute approximated Shapley values (Shapley, 1953) for each adapter via multilinear extension (Owen, 1972) on the visual texture classification task (Cimpoi et al., 2014) again for its representativeness and computational feasibility. As shown in Figure 8, vanilla LoRA exhibits skewed contributions, with 69% concentrated in the highest 4/12 Transformer layers. In contrast, LoRA-CoTo and Early LoRA-CoTo achieve more balanced contributions ($\pm 8\%$ and $\pm 3\%$ deviations across layers), confirming its efficacy in mitigating gradient imbalance.

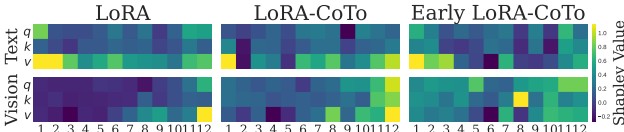

Figure 8. Approximated Shapley values of LoRA adapters by multilinear extension (Owen, 1972).

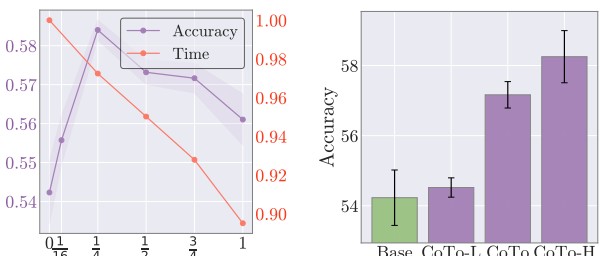

Figure 9. Ablation analysis of CoTo. Left panel: Impact of varying the proportion of training time allocated to the first (stochastic activation) phase on model accuracy (purple) and normalized training time (orange). Right panel: Comparison of dropout strategies, including no dropout (Base)), nested dropout from lower layers (CoTo-L), uniform dropout (CoTo), and nested dropout from higher layers (CoTo-H).

### 4.5. Ablation Studies

To systematically evaluate the design choices of CoTo and its robustness to hyperparameter variations, we conduct key ablation experiments on mathematical reasoning tasks (Cobbe et al., 2021). All experiments employ LoRA-Pro (Wang et al., 2024c) as the baseline.

**Training Phase Transition.** We first investigate the impact of varying the proportion of training time allocated to the stochastic activation phase (see left panel of Figure 9). The $x$-axis represents the percentage of total training spent in the first phase (where $p(t) < 1$), with 0% corresponding to vanilla LoRA (i.e., without CoTo) and 100% representing training exclusively with stochastic activation (i.e., $p(t)$ never reaches 1). Our results demonstrate that a 75% first-phase proportion strikes a good balance between task performance and training efficiency with secondary benefits like improved LMC.

**Dropout Strategy.** CoTo applies uniform dropout probability across all layers. To assess layer-specific effect, we design two variants: 1) CoTo-L, where adapters in lower layers are deactivated first, i.e., at time $t$, only adapters in layers $i > i_0$ (for some threshold $i_0$ determined by $p(t)$) remain active, and 2) CoTo-H, where adapters in higher layers are deactivated first, so that early in training only lower-layer adapters participate. The results reveal that CoTo and CoTo-H outperform CoTo-L, indicating that randomly deactivating—or prioritizing deactivation of—high-layer adapters is more effective. Nest-dropping from lower layers forces the model to rely prematurely on higher-layer

*Table 6.* Average accuracy (%) on mathematical reasoning tasks (Cobbe et al., 2021) for LoRA-Pro across different adapter ranks, learning rates, and insertion modules.

| LoRA Rank | 8 | 32 | 128 |
|---|---|---|---|
| LoRA-Pro | 54.23 ± 0.79 | 55.14 ± 1.73 | 56.48 ± 0.23 |
| + CoTo | 57.16 ± 0.38 | 57.24 ± 0.06 | **58.50** ± 0.46 |
| Learning Rate | 5e-5 | 1e-4 | 2e-4 |
| LoRA-Pro | 55.70 ± 0.96 | 55.85 ± 0.74 | 40.91 ± 1.09 |
| + CoTo | 55.83 ± 0.38 | **57.16** ± 0.38 | 56.25 ± 0.53 |
| Insertion Module | Attention | Projection | Gating |
| LoRA-Pro | 45.44 ± 0.40 | 49.08 ± 0.64 | 48.40 ± 0.16 |
| + CoTo | 52.41 ± 0.56 | **54.06** ± 0.80 | 51.96 ± 0.28 |

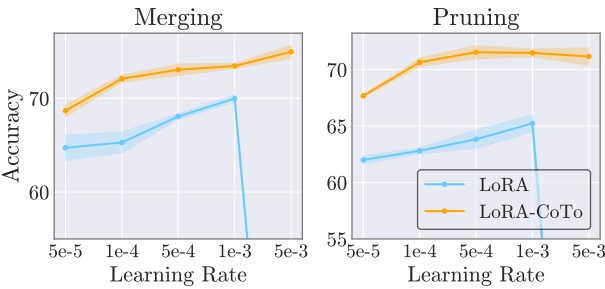

*Figure 10.* Average accuracy (%) on model merging and pruning for the visual texture classification task (Cimpoi et al., 2014). Left panel: Merging accuracy when $\lambda = 0.5$. Right panel: Pruning accuracy when removing alternating layers (*i.e.*, EveryOther).

adapters and undermines balanced optimization.

**Hyperparameter Sensitivity.** To demonstrate CoTo's compatibility with diverse LoRA configurations, we fine-tune LLaMA-2-7B using three adapter ranks ($r = 8, 32, 128$), three learning rates (5e-5, 1e-4, 2e-4), and three choices of insertion modules from attention, projection, and gating layers, respectively. From Table 6, we find that LoRA-Pro-CoTo consistently outperforms LoRA-Pro in all settings.

Further, Figure 10 extends this analysis to merging and pruning on the visual texture classification task (Cimpoi et al., 2014) across five learning rates. CoTo-trained adapters yield higher merging accuracy and maintain stronger robustness under structured pruning. These trends underscore CoTo's generality: it benefits LoRA variants across a wide spectrum of hyperparameter settings.

**Training Overhead Reduction.** Because CoTo stochastically deactivates adapters in early iterations, it reduces both forward and backward computation. We compare end-to-end fine-tuning times for LoRA, DoRA (Liu et al., 2024a), and HiRA (Huang et al., 2025), all under identical hardware and batch-size settings. From Table 7, we observe noticeable training overhead reduction, which arises because, when an adapter is inactive (*i.e.*, $\delta_i = 0$), its low-rank

*Table 7.* Wall-clock training times for LoRA, DoRA, and HiRA on mathematical reasoning tasks (Cobbe et al., 2021) using a single NVIDIA A6000 GPU.

| | LoRA | DoRA | HiRA |
|---|---|---|---|
| w/o CoTo | 7h 38min | 19h 00min | 11h 39min |
| w/ CoTo | 7h 05min | 14h 30min | 8h 50min |
| Speedup | 7.20% | 23.69% | 24.21% |

matrices are skipped entirely. Variants with larger adapter footprints (*e.g.*, DoRA and HiRA) thus experience more pronounced computational savings. Importantly, these gains accrue early in training yet do not compromise—or even improve—final accuracy.

## 5. Conclusion and Discussion

We have introduced CoTo, a progressive training strategy for LoRA. By gradually increasing adapter activation probability during training, CoTo promotes more balanced optimization across all layers while encouraging broader exploration of the loss landscape. Our theoretical analysis showed that CoTo enhances layer-wise dropout stability and LMC, while the cooperative-game perspective provided quantitative insights into each adapter's marginal contribution. Extensive experiments across vision-language models, large language models, and diffusion models consistently validated the effectiveness of CoTo.

While CoTo integrates seamlessly with diverse LoRA variants, a promising direction is to identify the optimal combination of existing LoRA "tricks." For instance, one could explore jointly applying CoTo's progressive schedule with adaptive-rank schemes (like AdaLoRA (Zhang et al., 2023b) or ALoRA (Liu et al., 2024b)), weight-decomposed updates (in DoRA (Liu et al., 2024a)), or Hadamard-based high-rank adaptations (in HiRA (Huang et al., 2025)), to determine how these techniques interact and where synergies arise. Systematically evaluating such combinations—potentially via automated hyperparameter search over activation schedules, rank allocations, and initialization strategies—could reveal configurations that maximize performance while minimizing parameter count and compute. Moreover, extending CoTo to jointly optimize over multiple PEFT objectives (*e.g.*, balancing dropout stability, quantization compatibility, and rank efficiency) could yield a unified framework that adapts to various resource constraints and task requirements.

## Acknowledgement

This work was supported in part by the National Key Research and Development Program of China (No. 2022ZD0160300) and the National Natural Science Foundation of China (No. 62136005).

## Impact Statement

This work aims to advance the field of machine learning. While it may carry various societal implications, none of which we feel must be highlighted here.

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

---

**Algorithm 1** CoTo

---

**Require:** Foundation model parameters $\boldsymbol{\theta}_0$, LoRA adapters $\Delta\boldsymbol{\theta} = \{\Delta\mathbf{W}_i\}_{i=1}^L$, and the total number of training steps $T$

1: **for** $t = 1, \ldots, T$ **do**
2:     Compute activation probability: $p = \min\left\{1, \frac{4t}{3T}\right\}$
3:     Draw a vector $\boldsymbol{\eta} = [\eta_1, \ldots, \eta_L]^\mathsf{T}$, where each $\eta_i$ is sampled independently from the uniform distribution $\mathcal{U}(0, 1)$
4:     **for** each layer $i = 1, \ldots, L$ **do**
5:         Set adapter state in layer $i$: $\delta_i = \mathbb{I}[\eta_i \leq p]$
6:     **end for**
7:     Compute the prediction $\hat{\boldsymbol{y}} = f\left(\boldsymbol{x}_0; \{\mathbf{W}_i + \delta_i \mathbf{1} \odot \Delta\mathbf{W}_i\}_{i=1}^L\right)$
8:     Compute the loss $\ell(\hat{\boldsymbol{y}}, \boldsymbol{y})$
9:     Compute the gradient $\nabla_{\{\Delta\mathbf{W}_i\}}\ell(\hat{\boldsymbol{y}}, \boldsymbol{y})$ and update the model parameters $\{\Delta\mathbf{W}_i\}$
10: **end for**

---

## A. Training Details

### A.1. Implementation Details

CoTo is implemented as a lightweight `TrainerCallback` within the PEFT ecosystem[4], enabling seamless integration with any LoRA-style fine-tuning loop. Algorithm 1 summarizes its computational procedure.

### A.2. Datasets

We evaluate CoTo across five computational prediction tasks: 1) image classification, 2) commonsense reasoning, 3) mathematical reasoning, 4) language understanding, and 5) image generation. For *image classification*, we follow Zanella & Ben Ayed (2024) and use 11 datasets:

- Aircraft (Maji et al., 2013) (aircraft classification)

- Caltech (Fei-Fei et al., 2007) (object recognition)

- Cars (Krause et al., 2013) (car classification)

- DTD (Cimpoi et al., 2014) (visual texture classification)

- EuroSAT (Helber et al., 2019) (satellite land classification)

- Flowers (Nilsback & Zisserman, 2008) (flower classification)

- Food (Bossard et al., 2014) (food classification)

- ImageNet (Deng et al., 2009) (large-scale object recognition)

- Pets (Parkhi et al., 2012) (pet breed classification)

- SUN (Xiao et al., 2010) (scene recognition)

- UCF (Soomro et al., 2012) (human action classification)

For *commonsense reasoning*, we use 8 tasks from Commonsense170K (Hu et al., 2023):

- ARC-c and ARC-e (Clark et al., 2018) (science questions)

- BoolQ (Clark et al., 2019) (yes/no questions)

- HellaSwag (Zellers et al., 2019) (commonsense inference)

- OBQA (Mihaylov et al., 2018) (multi-step reasoning)

---

[4]https://github.com/huggingface/peft

*Table 8.* Grid-searched learning rates for 11 image classification tasks (Zanella & Ben Ayed, 2024) across different LoRA variants.

| Method | Aircraft | Caltech | Cars | DTD | EuroSAT | Flowers | Food | ImageNet | Pets | SUN | UCF |
|---|---|---|---|---|---|---|---|---|---|---|---|
| LoRA | 2e-4 | 2e-4 | 2e-4 | 2e-4 | 2e-4 | 2e-4 | 2e-4 | 2e-4 | 2e-4 | 2e-4 | 2e-4 |
| LoRA-CoTo | 5e-4 | 5e-4 | 5e-4 | 5e-4 | 5e-4 | 5e-4 | 5e-4 | 5e-4 | 5e-4 | 5e-4 | 5e-4 |
| DoRA | 1e-3 | 1e-4 | 1e-3 | 1e-4 | 1e-4 | 1e-3 | 1e-4 | 1e-4 | 1e-4 | 1e-4 | 1e-4 |
| DoRA-CoTo | 1e-3 | 2e-4 | 1e-3 | 2e-4 | 1e-3 | 1e-3 | 2e-4 | 2e-4 | 2e-4 | 2e-4 | 2e-4 |
| HiRA | 5e-3 | 1e-3 | 5e-3 | 1e-3 | 5e-3 | 5e-3 | 1e-3 | 5e-3 | 1e-3 | 1e-3 | 1e-3 |
| HiRA-CoTo | 1e-2 | 5e-3 | 5e-2 | 5e-3 | 1e-2 | 5e-2 | 5e-3 | 5e-3 | 5e-3 | 5e-3 | 5e-3 |

*Table 9.* Standard deviation of classification accuracies across three random seeds for 11 image classification tasks.

| Method | Aircraft | Caltech | Cars | DTD | EuroSAT | Flowers | Food | ImageNet | Pets | SUN | UCF |
|---|---|---|---|---|---|---|---|---|---|---|---|
| LoRA | 0.75 | 0.12 | 0.24 | 0.97 | 0.81 | 0.19 | 0.22 | 0.06 | 0.27 | 0.18 | 0.44 |
| LoRA-CoTo | 0.72 | 0.10 | 0.29 | 0.39 | 0.44 | 0.36 | 0.18 | 0.04 | 0.39 | 0.20 | 0.22 |
| DoRA | 1.04 | 0.32 | 0.32 | 1.39 | 1.09 | 0.18 | 0.20 | 0.10 | 0.28 | 0.10 | 0.22 |
| DoRA-CoTo | 0.31 | 0.20 | 0.19 | 0.62 | 0.41 | 0.12 | 0.11 | 0.09 | 0.11 | 0.21 | 0.17 |
| HiRA | 0.27 | 0.09 | 0.25 | 1.33 | 1.33 | 0.02 | 0.09 | 0.14 | 0.09 | 0.11 | 0.23 |
| HiRA-CoTo | 0.15 | 0.11 | 0.08 | 0.28 | 0.05 | 0.15 | 0.12 | 0.10 | 0.36 | 0.13 | 0.35 |

- PIQA (Bisk et al., 2020) (physical commonsense reasoning)

- SIQA (Sap et al., 2019) (social reasoning)

- WinoGrande (Sakaguchi et al., 2021) (fill-in-the-blank questions)

For *mathematical reasoning*, we fine-tune on MetaMathQA (Yu et al., 2024) and test on GSM8K (Cobbe et al., 2021). For *language understanding*, we follow Zhao et al. (2024b) and use 9 tasks from GLUE (Wang et al., 2019) and Flan Collection (Longpre et al., 2023):

- CoLA (Dolan & Brockett, 2005) (linguistic acceptability)

- MNLI (Williams et al., 2018) (multi-genre natural language inference)

- MRPC (Dolan & Brockett, 2005) (paraphrase detection)

- QNLI (Rajpurkar et al., 2016) (question-answering)

- QQP[5] (Quora questions)

- RTE (Dagan et al., 2005; Bar Haim et al., 2006; Giampiccolo et al., 2007; Bentivogli et al., 2009) (textual entailment recognition)

- SNLI (Bowman et al., 2015) (natural language inference)

- SST2 (Socher et al., 2013) (sentiment analysis)

- WNLI (Levesque et al., 2012) (coreference resolution)

For *image generation*, we train on two content categories from DreamBooth (Ruiz et al., 2023) and two artistic styles from Hugging Face's LoRA the Explorer [6].

---

[5] https://data.quora.com/First-Quora-Dataset-Release-Question-Pairs
[6] https://huggingface.co/spaces/multimodalart/LoraTheExplorer

*Table 10.* LoRA hyperparameter configurations for different tasks. The dropout rate is predefined in the LoRAConfig class and applied independently of the proposed CoTo.

| Task | Rank | Default Learning Rate | CoTo Learning Rate | Insertion Module | Dropout Rate |
|------|------|----------------------|--------------------|--------------------|--------------|
| Image Classification | 2 | 5e-5 | 2e-4 | Attention | 0 |
| Commonsense Reasoning | 32 | 1e-5 | 5e-5 | Attention, Projection | 0.05 |
| Mathematical Reasoning | 8 | 2e-5 | 1e-4 | Attention, Projection, Gating | 0.1 |
| Language Understanding | 8 | / | 2e-5 | Attention ($\mathbf{Q}$, $\mathbf{V}$) | 0.05 |
| Image Generation | 16 | 1e-5 | 5e-5 | Attention | 0 |

*Table 11.* Average accuracy (%) on multi-task merging for 6 discriminative language understanding tasks (Wang et al., 2019) using the DeBERTa-v3 (He et al., 2023) backbone.

| | Learning Rate | Method | CoLA | MRPC | QNLI | QQP | RTE | SST2 | Avg |
|--|--------------|--------|------|------|------|-----|-----|------|-----|
| **w/o CoTo** | 5e-4 | LoRA | 87.44 | 89.46 | 94.31 | 91.05 | 85.56 | 95.18 | 90.50 |
| | | Fusion | 69.89 | 68.38 | 49.97 | 65.50 | 47.29 | 55.05 | 59.35 |
| | | Ensemble | 69.13 | 31.62 | 50.54 | 63.31 | 52.71 | 50.92 | 53.04 |
| | | LoRA-LEGO | 73.28 | 33.15 | 74.19 | 80.95 | 61.46 | 69.18 | 65.37 |
| | 1e-3 | LoRA | 86.48 | 88.48 | 93.87 | 91.16 | 84.12 | 94.95 | 89.84 |
| | | Fusion | 69.13 | 68.38 | 49.46 | 63.18 | 47.29 | 50.92 | 58.06 |
| | | Ensemble | 69.13 | 68.38 | 49.46 | 63.18 | 47.29 | 50.92 | 58.06 |
| | | LoRA-LEGO | 73.54 | 69.32 | 79.54 | 78.67 | 51.99 | 54.13 | 67.86 |
| **w/ CoTo** | 5e-4 | LoRA | 86.48 | 89.22 | 93.94 | 90.12 | 82.67 | 94.95 | 89.56 |
| | | Fusion | 69.22 | 68.38 | 67.12 | 70.32 | 47.29 | 58.14 | 63.41 |
| | | Ensemble | 70.66 | 40.69 | 51.31 | 66.71 | 47.29 | 66.40 | 57.18 |
| | | LoRA-LEGO | 53.75 | 70.77 | 76.19 | 75.26 | 64.31 | 89.01 | 71.55 |
| | 1e-3 | LoRA | 86.58 | 89.95 | 93.92 | 90.49 | 83.03 | 95.18 | 89.86 |
| | | Linear Fusion | 73.15 | 68.38 | 51.91 | 69.96 | 49.10 | 63.30 | 62.63 |
| | | Ensemble | 72.39 | 68.38 | 51.42 | 66.98 | 48.38 | 61.58 | 61.52 |
| | | LoRA-LEGO | 71.84 | 72.39 | 73.15 | 72.95 | 63.51 | 87.24 | 73.51 |

### A.3. Hyperparameters

We use publicly available implementations of CLIP-LoRA (Zanella & Ben Ayed, 2024), DoRA (Liu et al., 2024a), HiRA (Huang et al., 2025), LoRA-Pro (Wang et al., 2024c), LoRA-LEGO (Zhao et al., 2024b), and ZipLoRA (Shah et al., 2025), retaining original hyperparameters unless otherwise specified. Table 10 details the adapter rank, learning rate, insertion module, and dropout configurations for each task. Notably, HiRA requires a significantly higher learning rate (10–20× the default) for convergence. For *image classification*, we grid search learning rates around each method's default (see Table 8), while other tasks employ one initial learning rate paired with a cosine annealing schedule.

## B. Additional Experimental Results

### B.1. Image Classification

Standard deviations across 3 seeds remain low ($< 1.4\%$) for all image classification tasks (see Table 9), confirming result reliability in Table 1.

### B.2. Single-Task Merging for Commonsense Reasoning

Per-task interpolation curves (see Figure 11) show that CoTo consistently outperforms vanilla LoRA across all 8 commonsense reasoning tasks, with flatter loss basins.

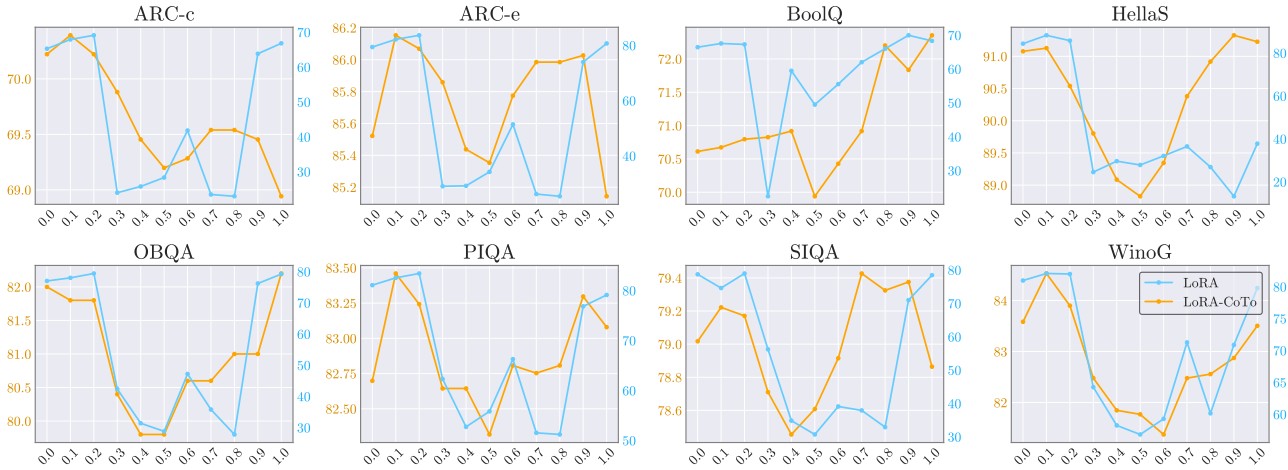

*Figure 11.* Linear interpolation accuracy on 8 individual commonsense reasoning tasks ([Hu et al., 2023](#)).

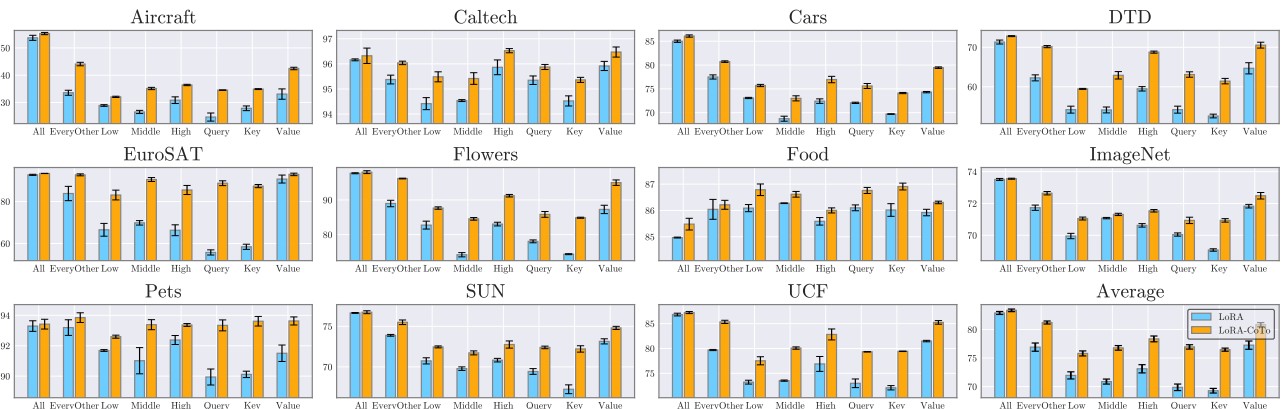

*Figure 12.* Average accuracy (%) on structured model pruning for 11 image classification tasks.

## B.3. Multi-Task Merging for Discriminative Language Understanding

CoTo improves merging accuracy by 4.18%–6.18% across linear weight fusion, linear model ensemble, and LoRA-LEGO ([Zhao et al., 2024b](#)) strategies (see Table 11), validating benefits in discriminative language tasks using the DeBERTa-v3 ([He et al., 2023](#)) backbone.

## B.4. Structured Pruning for Image Classification

CoTo maintains higher accuracy than vanilla LoRA under all structured pruning strategies across 11 image classification tasks (Figure 12).

# C. Extended Analysis

## C.1. Activation Schedule Ablation on DTD

Linear activation (CoTo default) balances convergence speed and robustness (see Figure 13). Exponential schedules improve merging/pruning but delay early convergence; sine schedules accelerate convergence but reduce robustness.

## C.2. Weight Convergence Visualization

t-SNE plots (see Figure 14) confirm LoRA adapters converge near initialization (*i.e.*, "lazy training") across learning rates. CoTo yields tighter clusters under initialization noise.

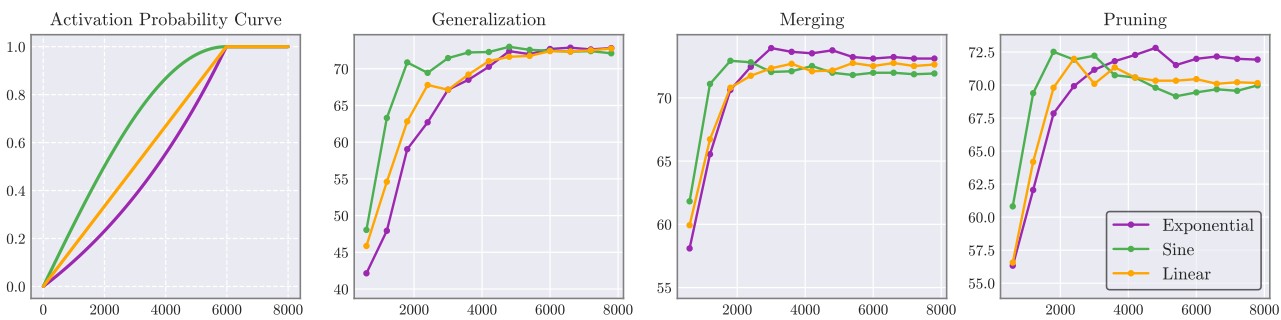

*Figure 13.* Performance evolution during training under different activation schedules on the visual texture classification task (Cimpoi et al., 2014). Merging accuracy is measured at $\lambda = 0.5$. Pruning accuracy is measured for $\mathrm{EveryOther}$ (*i.e.*, removing alternating layers).

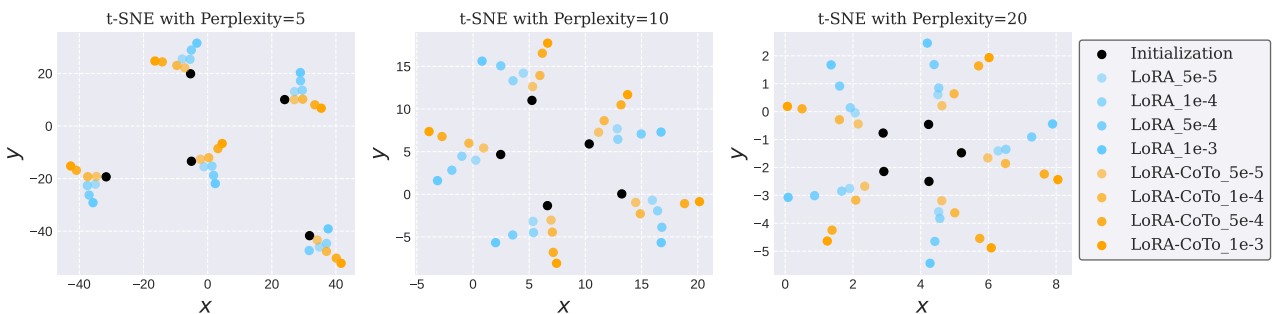

*Figure 14.* t-SNE visualization (Van der Maaten & Hinton, 2008) of the learned weights of LoRA adapters across five seeds and four learning rates, under three perplexity settings. Black dots denote initialization points, and color gradients indicate different learning rates.

## D. Proof of Theorem 3.1

**Theorem 3.1.** *Let $\ell(\cdot, \boldsymbol{y})$ be a convex loss function. Then for any fixed $p \in [0, 1]$,*

$$\min_{\{\Delta \mathbf{W}_i\}} \mathbb{E}_{\boldsymbol{\delta}} \left[ \ell \left( \hat{\boldsymbol{y}}, \boldsymbol{y} \right) \right] \geq \min_{\{\Delta \mathbf{W}_i\}} \sum_{j=1}^{L} w_j(p) \, \ell \left( \tilde{\boldsymbol{y}}_j, \boldsymbol{y} \right).$$

*Consequently, the expected CoTo objective at step $t$ upper-bounds a binomially weighted sum of the subnetwork losses.*

*Proof.*

$$
\begin{aligned}
\mathbb{E}_{\boldsymbol{\delta}} \left[ \ell \left( \hat{\boldsymbol{y}}, \boldsymbol{y} \right) \right] &= \mathbb{E}_{\boldsymbol{\delta}} \left[ \ell \left( f \left( \boldsymbol{x}_0; \{ \mathbf{W}_i + \delta_i \mathbf{1} \odot \Delta \mathbf{W}_i \}_{i=1}^{L} \right), \boldsymbol{y} \right) \right] \\
&= \sum_{j} \binom{L}{j} p^j (1-p)^{L-j} \mathbb{E}_{\|\boldsymbol{\delta}\|_1 = j} \left[ \ell \left( f \left( \boldsymbol{x}_0; \{ \mathbf{W}_i + \delta_i \mathbf{1} \odot \Delta \mathbf{W}_i \}_{i=1}^{L} \right), \boldsymbol{y} \right) \right] \\
&\geq \sum_{j} w_j(p) \ell \left( \mathbb{E}_{\|\boldsymbol{\delta}\|_1 = j} \left[ f \left( \boldsymbol{x}_0; \{ \mathbf{W}_i + \delta_i \mathbf{1} \odot \Delta \mathbf{W}_i \}_{i=1}^{L} \right) \right], \boldsymbol{y} \right) \\
&= \sum_{j} w_j(p) \ell \left( \tilde{\boldsymbol{y}}_j, \boldsymbol{y} \right).
\end{aligned}
\tag{8}
$$

$\square$

