# OpenReview forum: "Come Together, But Not Right Now: A Progressive Strategy to Boost Low-Rank Adaptation"
_ICML.cc/2025/Conference — ICML 2025 poster_

### Official Review · Reviewer_uABN · 2025-03-06

**Overall Recommendation:** 4

**Summary:**

The paper proposes a simple regularization strategy for LoRA fine-tuning that stochastically drops LoRA adapters according to a certain schedule.
The authors show that the proposed training strategy enhances linear mode connectivity (LMC) and adapter-wise dropout stability.
Furthermore, it usually improves fine-tuning over standard LoRA, DoRA and HiRA.
Finally, the authors present an analysis on robustness to pruning adapters, comparing linear fusion to ensembles, ablation studies on the dropout strategy and hyperparameters, using shapley values to determine importance of adapters, and custom generation with diffusion.

**Claims And Evidence:**

There are a few claims that lack support:

 - line 19 (right): "LoRA oftentimes rapidly overfits and converges to a local optimum near initialization." - no references or empirical evidence is presented
 - line 75 (right): The authors claim that PEFT follows a hierarchical structure - What is meant by hierarchical structure? Is it meant that higher layers need more adapter ranks? If so, the analysis in Section 5.2 showed that applying higher dropout in higher layer adapters performs better, which directly contradicts the claim that higher layers need more adapters. Also the analysis on Shapley values in Section 5.1 shows that for text, lower layer adapters are more important, indicating that there is also a dependence on data modality.
 - line 76 (right): The authors claim that CoTo accounts for the "hierarchical structure of PEFT" and is "tailored for PEFT", but no evidence is provided for that. In fact, the authors only consider LoRA-style methods, but PEFT comprises a lot more methods. Furthermore, the same dropout probability is applied to all adapter layers regardless of layer depth, i.e. adapters across layers are equally weighted.
 - In line 416 (left) the authors claim that 75% first stage proportion leads to improved LMC and dropout stability, however the authors have not compared e.g. LMC for 50% vs 75% first-stage, therefore this claim was not verified. It was only verified for the efficiency vs performance trade-off.
 - In line 430 (right), the authors claim that "Both theoretical and empirical analyses show that CoTo improves adapter-wise dropout stability and linear mode connectivity, leading to better generalization, interpolation, model merging, and pruning" - however, the theoretical results of this work only show that the CoTo objectiev is equivalent to a weighted sum of objectives, which does not tell anything about other findings, which are purely empirical.
 - In line 434 (right): "Shapley value analysis confirms that CoTo increases the marginal contribution of individual adapters." - This is not clear from looking at Figure 7, since not colorbars are shown and scale of different plots are unknown.

**Essential References Not Discussed:**

There is more related work [1,2,3] on adaptive rank allocation shows that for certain downstream tasks, naturally patterns emerge where more ranks are allocated to certain weights in higher layers and sometimes adapters are disregarded entirely. This could be used as additional support for dropping LoRA adapters stochastically during training.

[1] Liu et al., Alora: Allocating low-rank adaptation for fine-tuning large language models., NAACL 2024
[2] Paischer et al., One Initialization to Rule them All: Fine-Tuning via Explained Variance Adaptation, arXiv 2024
[3] Meo et al., Bayesian-LoRA: LoRA based Parameter Efficient Fine-Tuning using Optimal Quantization levels and Rank Values trough Differentiable Bayesian Gates. arXiv 2024

**Experimental Designs Or Analyses:**

I commend the authors for applying CoTo to different LoRA variants and the amount of experiments that were provided. However, I believe that a comparison to other stochastic regularization techniques such as Dropout [1] and Stochastic Depth, and [2] should be included. This forsters understanding of the training strategy and puts the proposed strategy into perspective to others.

[1] Lin et al., LoRA Dropout as a Sparsity Regularizer for Overfitting Control, arXiv 2024

[2] Wang et al., LoRA Meets Dropout under a Unified Framework, ACL 2024

The authors increase the learning rate for CoTo compared to competitors. In [1] it was shown that higher learning rate can be beneficial in the case of LoRA optimization. To enable a fair comparison to LoRA, I would recommend to also run LoRA with this same learning rate to account for that difference.

[1] Hayou et al., Efficient Low Rank Adaptation of Large Models, ICML 2024

The results on diffusion models and shapley values are entirely qualitatively. Based on this the authors claim significant improvements of CoTo over competitors, which is not convincing. In fact by merely looking at the generated images in Figure 5 it is very hard to tell which one is of better quality. The same goes for shapley values when no color bar is provided. I recommend to either remove claims on significant improvements or report quantitative measure with variance estimates to verify them.

Why does the selection of DTD for pruning analysis ensure generalizable insights?

On the multitask experiments, the authors use generative language models for language understanding tasks leading to very low scores in general. On such tasks BERT-style models such as DeBERTa-v3 [1] are the state-of-the-art usually reaching over 90% average accuracy. This model has also commonly been used in for LoRA-style fine-tuning [2,3,4], why do the authors use generative models here?

[1] He et al., DeBERTaV3: Improving DeBERTa using ELECTRA-Style Pre-Training with Gradient-Disentangled Embedding Sharing, ICLR 2023

[2] Hu et al., LoRA: Low-Rank Adaptation of Large Language Models, ICLR 2022

[3] Zhang et al., AdaLoRA: Adaptive Budget Allocation for Parameter-Efficient Fine-Tuning, ICLR 2023

[4] Meng et al., PiSSA: Principal Singular Values and Singular Vectors Adaptation of Large Language Models, NeurIPS 2024

In Section 4.2.1, the authors aim to answer the question whether CoTo makes linear fusion closer to an ensemble, but this question is never clearly answered. From my understanding the reported results only show that CoTo is usually better than LoRA*. To give a definitive answer I believe it needs to be looked at the magnitude of P for LoRA* vs LoRA-CoTo*, as if the magnitude of P is larger than there is a larger gap between ensemble and linear fusion.

**Methods And Evaluation Criteria:**

The proposed training strategy is very simple and straightforward which I consider a strength of the work.
It makes sense and stochastic regularization is an important topic for PEFT methods that has not been explored in a lot of detail.

**Other Comments Or Suggestions:**

Tables would be a bit nicer to read if subsequent rows were Method and Method-CoTo, separated by horizontal lines, otherwise one needs to jump back and forth across separated grouped rows.
The notion of adapters vs layers is a bit confusing. In the introduction the authors refer to LoRA adapters as "adapters", but then use "layers" throughout the remainder of the manuscript. This can be confused with a layer that contains multiple of adapters.
You could add a straight line in figure 4 that shows the pre-merge performance for LoRA, so the reader does not need to switch back and forth between table and plot.

-- POST REBUTTAL --

I would like to thank the authors for the added analysis which I believe improve the depth and rigor of the work. As all of my concerns have been properly addressed, I recommend acceptance of this work.

**Other Strengths And Weaknesses:**

**Strengths**

The proposed training strategy is simple and intuitive.

The authors conduct plenty of experiments and analyses demonstrating advantages of the proposed training strategy for different LoRA variants.

**Weaknesses**

The authors show learning curves for LoRA variants vs LoRA-CoTo variants in Figure 9 (Appendix B2) showing that CoTo requires longer training until convergence. This should also be mentioned in the main text to put the method in perspective.

In Table 4, Ensemble-CoTo performs consistently worse on average than the standard Ensemble method, do the authors have an intuition why this is the case?

**Questions For Authors:**

Have you looked at what P (used in LoRA*) learns during optimization? It seems that a good optimum would be to learn to invert one of the adapters for merging (i.e. inverting the fusion), such that eventually no fusion/ensembling is occuring at all.

No other remaining questions.

**Relation To Broader Scientific Literature:**

In general, there are plenty of methods that build on LoRA-style parameter-efficient fine-tuning.
Many of them investigate discarding more parameters, different initialization schemes, adaptive rank allocation, etc.
This paper investigates stochastic regularization for different LoRA variants, which provides a better understanding on how to best train LoRA-style methods.

**Theoretical Claims:**

Theorem 3.1 shows that training via CoTo is equivalent to a progressively shifted objective weighted by some factor.
I have not checked the proof for Theorem 3.1 in detail due to time constraints.

---

> ### Author Rebuttal · Authors · 2025-04-01
>
> We sincerely thank Reviewer uABN for the insightful feedback, which helped improve both our analysis and presentation. Below, we respond to the main concerns point by point. Additional results are available via https://anonymous.4open.science/r/coto, with new content labeled as Tab. rX and Fig. rX.
>
> #### Claims And Evidence
>
> > We agree with these concerns and will revise the claims for clarity.
> >
> > - L19,r: Fig. r4 shows that models with the same initialization tend to converge to similar solutions, regardless of method or learning rate. We will also cite works supporting lazy training dynamics.
> >
> > - L75-76,r: The term “hierarchical structure” was unclear. We will revise it to *layer-wise importance differences*, note that layer importance is modality-dependent, and specify “layer-wise LoRA-style PEFT methods” instead of the broader term PEFT.
> >
> > - L416,l: Fig. 6-7 in the manuscript show pruning and marginal contributions at 25% training. Fig. r1 provides additional merging and pruning results over training steps.
> >
> > - L430,r: As shown in [r1], the dropout objective is equivalent to adding a regularization term. Similarly, CoTo’s objective is a weighted sum of sub-objectives, where the weight ($w_6$ in Fig.2) of the standard objective gradually dominates as training progresses.
> >
> >   [r1] On the Regularization Properties of Structured Dropout.
> >
> > - L434,r: Fig. r5 has been updated with unified colorbars.
>
> #### Experimental Designs
>
> [E.1] Comparison to other stochastic techniques
>
> > Stochastic depth is designed for pretraining and not applicable to LoRA. [2] introduces a KL loss between dropout and non-dropout outputs during PEFT but is not open-sourced or directly applicable. Thus, we compare CoTo with Dropout [1] and a variant using our progressive strategy. As shown in Fig. r2 and Fig. r3, CoTo consistently outperforms naive dropout in general, merging and pruning tasks.
>
> [E.2] Run LoRA with the same learning rate
>
> > - In Tab. 5 of ablation studies, we add LoRA-Pro results with 1e-4 and 2e-4 for fair comparison (55.8 ± 0.7, 40.9 ± 1.1). On classification tasks, CoTo also consistently outperforms LoRA on general, merging, and pruning performance across five learning rates (Fig. r6).
> > - The higher learning rate for CoTo offsets slower convergence from stochastic training. Full learning rate settings are provided in Appendix A.3.
>
> [E.3] Remove claims or report quantitative measures
>
> > We will remove “significant” and move diffusion results to Appendix. Additional quantitative results (Tab. r2) and qualitative examples will be included.
>
> [E.4] Why DTD for pruning analysis
>
> > DTD has relatively low zero-shot performance (~44%), making the LoRA's gain more visible. Results on all vision tasks are provided in **Fig. r7** for completeness.
>
> [E.5] Why use generative models instead of BERT-style models like DeBERTa-v3
>
> > - We follow LoRA-LEGO, which evaluates only generative LLMs, as they are more advanced and suitable for merging. BERT models for classification rely on task-specific poolers and classifiers, making merging less meaningful (e.g., not generalizable to OOD tasks). In contrast, generative models support instruction tuning across diverse tasks via a unified prompt format [r2], offering a more rigorous testbed.
> >
> >   [r2] Finetuned Language Models are Zero-Shot Learners.
> >
> > - As suggested, we have also added results on DeBERTa-v3 in Tab. r1, where CoTo still shows consistent improvements.
>
> [E.6] Analysis of the magnitude of P
>
> > We optimize the alignment matrix P by minimizing the proposed $\Delta_{upper}$ using Adam (500 steps, lr=0.01) and analyze both the magnitude of P and the difference $\|\Delta W_f - \Delta W_m\|_2$ before and after applying P. Results are shown in Fig. r8. For LoRA, the magnitude of P is larger, and the difference (both before and after applying P) is smaller than in CoTo. This indicates that CoTo improves linear mode connectivity not by adapter-wise alignment but by addressing layer-wise misalignment.
>
> #### Essential References
>
> > Thanks for highlighting these works. We will cite and discuss them in the revision.
>
> #### Weaknesses
>
> [W.1] Fig. 9 shows that CoTo requires longer training until convergence
>
> > The slower loss decrease in Fig. 9 is due to CoTo’s stochastic activation, which delays early convergence but improves generalization. Notably, CoTo with LoRA and DoRA reaches lower loss at 900 and 2k steps, respectively. Also, as each step in CoTo is faster (due to adapter skipping), total training time is reduced (Tab. 6). Any delay from stochasticity can be mitigated by adjusting the learning rate or activation schedule (Fig. r1).
>
> [W.2] Intuition why Ensemble-CoTo performs worse
>
> > Ensemble-CoTo improves on DeBERTa (Tab. r1) but underperforms on LLaMA, possibly due to non-convergence. We will further investigate this and update the results if new findings emerge.
>
> #### Comments
> > Thanks, we will revise accordingly.
>
> #### Questions
> > Please see our response to E.6.

---

> > ### Comment · Reviewer_uABN · 2025-04-03
> >
> > Thank you for addressing most of my comments and adding additional very insightful results, I have a few more remaining questions and comments.
> >
> > > L19,r: Fig. r4 shows that models with the same initialization tend to converge to similar solutions, regardless of method or learning rate. We will also cite works supporting lazy training dynamics.
> >
> > The original claim was that "adapters converge to a local optimum near initialization", Fig r4 shows that same initialization tend to converge to similar solutions, however what is not supported is that the reached optimum is close to initialization. To verify this, you would need to compute the distance between initialization and trained adapter and this should be relatively small, which is what I doubt. Another remak is that tSNE projections are very sensitive to hyperparameters, so they usually dont tell much.
> >
> > > Stochastic depth is designed for pretraining and not applicable to LoRA. [2]
> >
> > Can you elaborate why it is not applicable? Just because it is designed for pretraining does not mean it cannot be applied to fine-tuning.
> >
> > > Why use generative models instead of BERT-style models like DeBERTa-v3
> >
> > I appreciate adding results for DeBERTav3. It seems that LoRA without CoTo consistently outperforms LoRA with CoTo, while merging performance still improves. I agree though that investigating the effect of merging on OOD tasks is not as meaningful with such models compared to generative ones, still they hold sota performance in classification. I am satisfied with these results.
> >
> > If the authors can address the first to points properly, I will increase my score.

---

> > > ### Author Response · Authors · 2025-04-05
> > >
> > > We are deeply grateful to the reviewer for the thoughtful follow-up and the opportunity to clarify the remaining concerns. As part of our response, we have added new results in Fig. r9 and Tab. r3.
> > >
> > > ### Validity of the “Convergence Near Initialization” Claim
> > > > We appreciate the reviewer for rigorously examining every claim in our paper. To further substantiate our original claim that "adapters converge to a local optimum near initialization", we have
> > > > - replicated the analysis from Fig. r4 five times using different initializations and reported the following table, which summarizes the **average $\ell_2$ distance between the initialized and final LoRA adapter weights** across learning rates. Each value is averaged over 5 seeds. There are some key observations:
> > > >     - For both LoRA and LoRA-CoTo, **the final weights remain much closer to their initialization** than to weights trained from different seeds, even at initialization, supporting convergence near initialization in line with lazy training dynamics.
> > > >     - LoRA-CoTo shows slightly larger distances from initialization, indicating **broader exploration** during training, yet its final weights are more consistent across seeds, suggesting a more **consistent convergence path**.
> > > >   - While we agree with the reviewer that t-SNE visualization is sensitive to hyperparameters, t-SNE that has been widely adopted for cross-checking clusters corroborates our claim. As shown in Fig. r9 (an updated version of Fig. r4 with varying perplexities), the final adapters form five distinct clusters centered around their initializations.
> > > >
> > > >    | Method| Comparison|5e-5|1e-4 (default) |5e-4|1e-3| Note|
> > > >    | --- | --- | --- | --- | --- | --- | --- |
> > > >    | Both| Init vs. Init (diff. seeds)|1.155±0.002| 1.155±0.002| 1.155±0.002 | 1.155±0.002 ||
> > > >    | LoRA| Init vs. Final (same seed)|0.476±0.016 |0.445±0.018|0.757±0.010| 1.315±0.015 ||
> > > >    | LoRA-CoTo | Init vs. Final (same seed)|0.610±0.004 |0.789±0.004| 1.251±0.024| 1.637±0.025 |Slightly larger |
> > > >    | LoRA| Final vs. Final (diff. seeds) |1.703±0.031| 1.810±0.035|2.348±0.030| 3.117±0.027 ||
> > > >    | LoRA-CoTo | Final vs. Final (diff. seeds) |1.380±0.004| 1.533±0.006|2.142±0.030 | 2.631±0.022 |Slightly smaller |
> > > >
> > > > - added an analysis where we **perturb the initialization** in a very small range by adding small uniform noise to each adapter and examine **whether the adapted LoRAs converge within this range**. For three seeds with three perturbations each, we compute the average $\ell_2$ distance between final weights. The results below show that when initialization points are close, the **final weights also remain tightly clustered**, further supporting the claim of convergence near initialization.
> > > >
> > > >   | Method| Comparison (diff. perturbations) |5e-5| 1e-4| 5e-4|1e-3|
> > > >   | --- | --- | --- | --- | --- | --- |
> > > >   | Both| Init vs. Init |0.020±0.000 |0.020±0.000 |0.020±0.000 |0.020±0.000 |
> > > >   | LoRA| Final vs. Final |0.049±0.005 |0.065±0.006|0.593±0.016|1.712±0.014|
> > > >   | LoRA-CoTo|Final vs. Final |0.040±0.009 |0.055±0.004| 0.208±0.007|0.515±0.029|
> > >
> > > ### Applicability of Stochastic Depth in the Context of LoRA Fine-tuning
> > > > We thank the reviewer for always inspiring us to dig further and make our results more solid.
> > > > - We would like to humbly clarify that our intended message was that Stochastic depth designed for pre-training skips entire Transformer layers—namely, it operates at the **inherent Transformer parameters**—is inconsistent with our fine-tuning setup, which tunes **adapters only**. We apologize for any confusion.
> > > > - Part of our core contributions, i.e., (1) identifying the deficiencies of current LoRA optimization with uneven distribution across layers and (2) proposing a progressive training strategy on adapters to address such deficiencies, is exactly the bridge to close such inconsistency.
> > > > - Based on our above contributions, we have followed the reviewer's thought to transfer the structured, layer-wise dropout strategy of stochastic depth to our setting.
> > > >   - We apply the same linear decay schedule across adapter layers as in Stochastic depth, i.e., a linear decay in activation probability from $p = 1$ (first layer) to $p = 0.5$ (last layer). We denote this variant as `Stochastic Depth for LoRA`, where early adapter layers are always active while later adapter layers are occasionally skipped.
> > > >   - We have compared CoTo against `Stochastic Depth for LoRA` on *DTD* and *UCF101* classification tasks and summarized the results in Tab.r3.
> > > >     - `Stochastic Depth for LoRA` indeed improves merging and pruning, confirming that structured adapter skipping is indeed effective—an insight shared by CoTo.
> > > >     - CoTo still outperforms `Stochastic Depth for LoRA` consistently across generalization, merging, and pruning, and it goes beyond stochastic depth by introducing a **progressive activation schedule** over training steps, which stabilizes layer-wise optimization and enhances generalization.

---

### Official Review · Reviewer_Xi9N · 2025-03-09

**Overall Recommendation:** 3

**Summary:**

The paper introduces CoTo, which integrates structured dropout with LoRA fine-tuning, demonstrating improved generalization and enhanced performance in model merging and pruning. Similar to stochastic depth, the proposed method freeze LoRA adaptor for certain layers with a certain probability and such probability decreases over the entire training process.

**Claims And Evidence:**

Yes.

**Essential References Not Discussed:**

No.

**Experimental Designs Or Analyses:**

I have checked the validity of the experimental design. I believe it is natural to compare LoRA-variants without CoTo and those with CoTo.

**Methods And Evaluation Criteria:**

The proposed method is simple and not new. It is very intuitive and makes sense for improving generalization.

**Other Comments Or Suggestions:**

While the overall novelty of the paper is somewhat limited, i.e., mainly combining existing techniques such as stochastic depth and structured dropout within the PEFT framework, the strength of the experimental results helps compensate for this. The experiments are extensive, and demonstrate significant improvements over baselines. Therefore, I am inclined to recommend acceptance of the paper.

**Other Strengths And Weaknesses:**

Strengths:
1. The proposed method shows consistent improvements compared to baselines.
2. The experiments are extensive, including VLM, and LLM.
3. The paper also provides extensive studies, including how CoTo affects model merging, and pruning. It provides interesting insights to the researchers in the field.

Weaknesses:
1. As the authors mentioned in related works, there are many works that tested applying structured dropout and stochastic depth. The novelty of the paper is limited, i.e., applying structured dropout to PEFT methods.

**Questions For Authors:**

I do not have other questions.

**Relation To Broader Scientific Literature:**

Researchers in this field may get interested in the paper as it improves generalization of PEFT methods.

**Theoretical Claims:**

I did not check the correctness of Theorem 3.1.

---

> ### Author Rebuttal · Authors · 2025-04-01
>
> We sincerely thank Reviewer Xi9N for the thoughtful review and for highlighting both the strengths and limitations of our work. We especially appreciate the recognition of our experimental rigor and the inclination toward acceptance. Below, please find our responses to the main concerns, and let us know if any issues remain. Additional experiments conducted during the response period (see Fig. **r**X & Tab. **r**X) are available in this anonymous link: https://anonymous.4open.science/r/coto.
>
>
> #### [Weakness] As the authors mentioned in related works, there are many works that tested applying structured dropout and stochastic depth. The novelty of the paper is limited, i.e., applying structured dropout to PEFT methods.
>
> > We fully understand the concern regarding novelty. Indeed, from a structural perspective, **our method may appear similar to prior work** on structured dropout and stochastic depth. We acknowledge this and appreciate the reviewer’s valuable observation.
> >
> > - We welcome the opportunity to clarify CoTo's distinct contributions and novelty within the PEFT landscape. While there are superficial similarities, CoTo's core innovation goes beyond simply applying existing ideas. As discussed in **Appendix C** and summarized in **Table 8**, while CoTo, dropout, and stochastic depth all involve deactivation during training, they differ fundamentally in objective:
> >
> >   - **Dropout** aims to prevent overfitting by randomly deactivating individual neurons or weights.
> >
> >   - **Stochastic depth** is primarily designed for efficient pretraining of full-model by randomly skipping entire layers.
> >
> >   - **CoTo** is designed as a training paradigm for the **layer-wise PEFT method**, aiming to **balance the utilization of LoRA adapter layers**. It prevents adapters from dominating and promotes better merging and pruning by employing **structured, layer-wise deactivation coupled with a progressive activation strategy**.
> >
> >   In essence, CoTo synthesizes concepts from regularization (like dropout) and efficient training (like stochastic depth) but repositions and adapts them into a coherent strategy specifically targeting the unique dynamics and optimization challenges of layerwise PEFT.
> >
> > - To empirically demonstrate this distinction, we provide additional experiments (**Anonymous Figs. r2 and r3**) on image classification comparing CoTo against direct applications of dropout to LoRA adapters:
> >
> >   - Standard Dropout (fixed $p$).
> >
> >   - Progressive Dropout (linearly decaying $p$).
> >
> >   **Fig. r2** shows LoRA adapter merging robustness under multiple seeds, and **Fig. r3** evaluates structured pruning across adapter components. The results clearly show that **naive applications of dropout strategies yield limited benefits** and do not effectively improve merging or pruning robustness. CoTo consistently and significantly outperforms these variants in terms of accuracy, merging performance, and pruning robustness, highlighting that its effectiveness stems from its PEFT-specific design, not merely from applying dropout principles.
> >
> > Finally, we note that **CoTo is fully compatible with standard dropout**, which is already included in our main experiments. We will revise the manuscript to better highlight these conceptual and empirical distinctions, helping readers better position CoTo within the broader landscape of PEFT techniques.

---

> > ### Comment · Reviewer_Xi9N · 2025-04-04
> >
> > Thank you for your rebuttal. I am keeping my original rating.

---

### Official Review · Reviewer_VYdq · 2025-03-13

**Overall Recommendation:** 3

**Summary:**

The paper introduces a training strategy to progressively deactivate adapters during training to ensure better optimization across all layers, enhancing model performance and efficiency. Extensive experiments across various models and tasks demonstrate its effectiveness in boosting LoRA's capabilities, including improved generalization, better model merging, and efficient pruning.

**Claims And Evidence:**

Yes, the claims made in the submission are supported by clear and convincing evidence.

**Essential References Not Discussed:**

Please refer to Strengths and Weaknesses.

**Experimental Designs Or Analyses:**

Yes, I checked the soundness of all experimental designs.

**Methods And Evaluation Criteria:**

Yes, the proposed method and evaluation criteria make sense for the problem.

**Other Comments Or Suggestions:**

Please refer to Strengths and Weaknesses.

**Other Strengths And Weaknesses:**

Strengths:

1. CoTo provides a novel approach to addressing the uneven distribution of LoRA updates across layers.
2. The strategy is supported by both theoretical insights and empirical evaluations, showcasing consistent performance improvements.
3. CoTo is compatible with various LoRA variants and extends its benefits to tasks such as model merging and pruning, demonstrating broad applicability.

Weaknesses:

1. The paper's notation is a bit unclear. What exactly does "a single LoRA layer" refer to? And what does "adapter" refer to? In transformer models, a single layer typically contains multiple weight matrices, such as the attention weight matrix and the projection weight matrix. Does "one adapter" mean all the LoRA modules added to these matrices in one layer?
2. Following up on the previous question, could you explain in more detail how the adapter drop method works?
3. In Table 1, the improvement in performance seems marginal. How do the experiments show that CoTo can enhance generalization? Are any held-out tasks set aside to test this?
4. Line 160-164, why does a decreasing learning rates facilitates the exploration of new local optima? Is this an assumption? What's the strategy for decreasing the learning rate?

**Questions For Authors:**

Please refer to Strengths and Weaknesses.

**Relation To Broader Scientific Literature:**

Please refer to Strengths and Weaknesses.

**Theoretical Claims:**

Yes, I checked the correctness of the proofs for theoretical claims.

---

> ### Author Rebuttal · Authors · 2025-04-01
>
> We sincerely thank Reviewer VYdq for the thoughtful and constructive comments. We address each concern in detail below and remain open to any further questions.
>
> [W1] Notation is a bit unclear. What exactly does "a single LoRA layer" refer to? And what does "adapter" refer to? Does "one adapter" mean all LoRA modules added to these matrices in one layer?
> > We greatly appreciate the reviewer for bringing our attention to this notation, which indeed benefits from further clarification.
> >
> > - Adapter: As stated in Lines 11–13 (right) of our manuscript, we use the term adapter to refer to the trainable parameters introduced by LoRA.
> > - A single LoRA layer = a single adapter denotes **the collection of all LoRA modules inserted into a single Transformer layer**.
> >   - The exact inserting locations of LoRA within each layer vary by task, following state-of-the-art practices, as detailed in the "Target Module" of Tab. 7.
> >   - Consequently, activation decisions are applied collectively to all LoRA modules within a layer, rather than to individual modules.
>
> [W2] Explain in more detail how the adapter drop method works
> >Absolutely. Thank you again for the question. The adapter drop mechanism is a key component of our training strategy, described in Sec. 3.1 and illustrated in **Fig. 1**, and we appreciate the opportunity to clarify it further.
> >
> >At each training step, we sample a Bernoulli variable $\delta_l\sim \text{B}(p)$ for each layer to determine **whether the entire layer of adapters is activated**. This sampling is implemented using the `TrainerCallback` function at each step. The activation probability $p$ increases linearly from 0 to 1 during the first three-quarters of training and remains fixed at 1 thereafter. When deactivated, **all LoRA modules within a transformer layer** are excluded from both forward and backward computation. As shown in Fig. 1, this strategy enables extensive layer-skipping during the early stages of training and gradually transitions to full activation of all adapters.
>
> [W3] Table 1 performance improvement seems marginal
> > Thank you for this insightful question. While the improvements in Tab. 1 may appear modest in absolute terms, we note that:
> >
> > - Tables 1 and 2 report results on standard benchmarks, where performance is already strong. CoTo still achieves consistent gains while maintaining lower cost (Sec. 5.3). Compared to prior methods, such as DoRA vs. LoRA and HiRA vs. DoRA, CoTo provides comparable or even greater improvements, setting new state-of-the-art results in several settings.
> > - In addition to generalization accuracy gains, CoTo significantly enhances the **merging and pruning performance** of LoRA adapters. This is another major benefit of our method.
> >
> > CoTo improves generalization at both the **sample level** and the **task level**:
> >
> > - Tab. 1–3 evaluate performance on the test sets of in-domain tasks, demonstrating sample-level generalization. In Tab. 3, we train on MetaMathQA and test on GSM8K, which differ in domain, reflecting robustness under domain shift.
> > - Tab. 4 includes explicitly **held-out tasks** in the merging experiments. Following the LoRA-LEGO protocol, we train LoRA adapters on a subset of tasks (in-domain) and evaluate merged adapters on both seen (ID) and unseen (OOD) tasks. CoTo consistently improves performance on these held-out tasks, demonstrating strong task-level generalization.
> >
> > We will revise the manuscript to more clearly highlight these generalization settings and the role of held-out tasks in our experimental design.
>
> [W4] Line 160-164, why does a decreasing learning rates facilitates the exploration of new local optima...
> >  We acknowledge that the original statement "Coupled with decreasing learning rates, this strategy facilitates the exploration of new local optima while preserving properties established in earlier stages" was confusing and appreciate the opportunity to clarify.
> >- The "strategy" referenced here is the proposed CoTo strategy, which is responsible for facilitating the exploration of new local optima.
> >- What we intended to deliver is that "decreasing learning rates" coupled with the lazy training property of neural networks (cf. Line 158 - 160) is **responsible for "preserving properties established in earlier stages"**, such as linear mode connectivity and dropout stability.
> >   - Specifically, as supported by prior work [r1, r2], using a higher learning rate during the early training phase promotes broad exploration of the loss landscape. Subsequently, gradual learning rate decay encourages convergence to flatter, more generalizable minima. In our empirical setting, we adopt a cosine decay learning rate schedule, which we believe helps maintain early-stage properties.
> >  We will revise the manuscript to better reflect this intent.
> >
> >  [r1] SGDR: Stochastic Gradient Descent with Warm Restarts.
> >
> >  [r2] A Second look at Exponential and Cosine Step Sizes: Simplicity, Adaptivity, and Performance.

---

> > ### Comment · Reviewer_VYdq · 2025-04-06
> >
> > I thank the authors for the detailed response. Most of my concerns are addressed. After considering the insights from other reviewers, I increased my score to 3.

---

### Official Review · Reviewer_Np5j · 2025-03-13

**Overall Recommendation:** 2

**Summary:**

This paper proposes CoTo, a training strategy for LoRA that progressively deactivates adapters during training to promote balanced optimization across layers. CoTo enhances generalization, model merging, and pruning while reducing training time, demonstrating performance improvements across vision-language models (CLIP), LLMs (LLaMA), and diffusion models (SDXL).


## update after rebuttal

I am still not convinced by the reply to Q2 regarding the theoretical analysis. I am keeping my score unchanged.

**Claims And Evidence:**

Most claims are well-supported by empirical evidence, including performance gains, improved model merging, and pruning efficiency across multiple benchmarks. However, the theoretical analysis is limited to fully connected networks, and layer-wise optimization balance lacks direct gradient-based validation.

**Essential References Not Discussed:**

NA

**Experimental Designs Or Analyses:**

The experiments cover various models and LoRA variants, with clear results supporting CoTo's improvements. However, layer-wise optimization balance is not directly validated using gradient-based metrics.

**Methods And Evaluation Criteria:**

The methodology and evaluation use diverse benchmarks across vision-language models, LLMs, and diffusion models. However, layer-wise optimization balance lacks gradient-based validation, and the choice of a linear activation schedule is not compared to alternatives.

**Other Comments Or Suggestions:**

NA

**Other Strengths And Weaknesses:**

Lack of Mathematical Analysis on Gradient Distribution and Convergence: The paper does not provide a detailed mathematical analysis of gradient distribution or training convergence, leaving uncertainty about the optimization stability and convergence behavior of CoTo.

Narrow Scope of Adapter Merging Tests: The experiments on CoTo's adapter merging are limited to LLaMA-2 and LLaMA-3. Existing works highlight the importance of adapter merging across different tasks/domains, but this paper's scope is too narrow to generalize.

**Questions For Authors:**

Why use a linear increase in activation probability?

**Relation To Broader Scientific Literature:**

The paper relates to the PEFT literature, building on LoRA and its variants while introducing stochastic adapter activation to improve layer-wise optimization. It also connects to Linear Mode Connectivity (LMC) for model merging and extends dropout and stochastic depth methods to adaptively adjust activation for balanced optimization.

**Theoretical Claims:**

The theoretical analysis may not fully generalize to transformers in attention layers.

---

> ### Author Rebuttal · Authors · 2025-04-01
>
> We sincerely thank Reviewer Np5j for the valuable comments. Below, please find our responses to each concern, and let us know if any issues remain. All experiments during the response period (Fig. **r**X & Tab. **r**X) are accessible in this anonymous link https://anonymous.4open.science/r/coto.
>
> #### [Q1]  Direct gradient-based validation for layer-wise optimization balance
>
> > - **Gradient-based validation**
> >
> >   To directly validate layer-wise optimization balance, we compute the **average gradient magnitudes of layers** from both the text and vision encoders on vision classification tasks. For comparability, gradients are normalized such that their sum across layers equals one. The results below indicate that CoTo leads to a **more balanced gradient distribution** across layers:
> >
> >   |                    | Lower Layers (0-4) | Middle Layers (5-8) | Higher Layers (9-12) |
> >   | ------------------ | ------------------ | ------------------- | -------------------- |
> >   | LoRA (text)        | 27.59%             | 32.35%              | 40.06%               |
> >   | LoRA-CoTo (text)   | 28.48%             | 33.53%              | 37.99%               |
> >   | LoRA (vision)      | 24.06%             | 32.59%              | 43.35%               |
> >   | LoRA-CoTo (vision) | 26.74%             | 33.90%              | 39.35%               |
> >
> > - **Other validations already presented in our manuscript**
> >   - Fig. 6 shows that CoTo-trained models consistently outperform LoRA, regardless of whether the lower, middle, or higher layer-adapters are pruned.
> >   - Fig. 7 presents Shapley value analyses, revealing that CoTo enhances the marginal contributions of lower and middle layers.
> >
> > Together, these results support the effectiveness of CoTo in promoting balanced layer-wise optimization.
>
>
> #### [Q2]  The theoretical analysis may not fully generalize to transformers in attention layers.
> > For clarity and simplicity of notation, we formulate our method using fully connected networks, where $W_l$ represents the total parameters of layer $l$. However, since our analysis (e.g., Theorem 3.1) considers activation at the layer level (as described in Sec. 3.1), it does not rely on the internal structure of each layer. By the Law of Large Numbers, the CoTo objective can be reasonably approximated as a weighted sum of sub-objectives, making the analysis broadly applicable — including to transformer-based architectures.
>
>
> #### [Q3]  The experiments on CoTo's adapter merging are limited to LLaMA-2 and LLaMA-3. Existing works highlight the importance of adapter merging across different tasks/domains, but this paper's scope is too narrow to generalize.
>
> > We respectfully clarify below the **broad scope** of adapter merging evaluations, which we have deliberately designed to validate CoTo's high generalizability.
> > - **Architectural scope includes LLaMA-2/LLaMA-3/CLIP/SDXL/DeBERTa V3**
> >    - As recognized by your summary, we evaluate adapter merging on CLIP (Fig. 4), LLaMAs (Fig.4, Tab. 4), and SDXL (Fig. 5), covering **widely adopted architectures** in text, vision, and multimodal domains.
> >    - We also evaluate CoTo on the **BERT-style architecture** DeBERTa V3 (Tab. r1). Despite the challenges of merging BERT-style models -- requiring additional fine-tuning of task-specific poolers and classifiers -- CoTo still outperforms.
> > - We fully agree that adapter merging across different tasks/domains is critical. Aware of this, Sec. 4.2.2 ("LoRA Merging with Different Tasks") merges 7 adapters trained on 7 in-domain (ID) tasks, and evaluates the cross-task merged adapter on both ID and 2 out-of-domain (OOD) tasks. The consistent superiority with CoTo over baselines advocates our **broad scope in cross-task adapter merging**.
> > - Besides, our adapter-merging evaluations span diverse task types, including vision classification (Fig. 4), commonsense reasoning (Fig. 3), natural language understanding (Tab. 4), and customized generation via diffusion models (Fig. 5), further validating **CoTo's generalizability across tasks/domains.**
>
>
> #### [Q4]  Why use a linear increase in activation probability?
>
> > We select linear increase as it offers a **favorable trade-off between simplicity and performance**, preserving the core benefits of the proposed CoTo.
> > - Simplicity: Unlike other activation probability scheduling functions (e.g., sine and exponential), linear increase **requires no additional hyperparameters** given the first phase spanning 75% of the training duration (cf. Line 410 (left)).
> > - Performance: Following the reviewer's great suggestion, we compare **linear**, **sine**, and **exponential** activation probability functions. As evidenced in Fig. r1, linear increase strikes a delicate balance between generalization, merging, and pruning performance.

---

### Decision · Program_Chairs · 2025-05-01

**Decision:**

Accept (poster)

**Comment:**

The paper develops an interesting stochastic approach to activate/deactivate adapters during training to improve low-rank adaptation (LoRA). The reviewers generally agree that the paper does an extensive set of experiments, and evaluates the ideas on all the relevant metrics. One of the reviews is more negative, but the issues do not seem to be serious enough.